# A Continuized View on Nesterov Acceleration for Stochastic Gradient Descent and Randomized Gossip

**Mathieu Even[1],[*], Raphaël Berthier[1],[*], Francis Bach[1], Nicolas Flammarion[2], Pierre Gaillard[3], Hadrien Hendrikx[1], Laurent Massoulié[1],[4] and Adrien Taylor[1]**

[*] Equal contributions

[1]Inria - Département d'informatique de l'ENS, PSL Research University, Paris, France

[2]School of Computer and Communication Sciences
Ecole Polytechnique Fédérale de Lausanne

[3]Univ. Grenoble Alpes, Inria, CNRS, Grenoble INP, LJK, 38000 Grenoble, France

[4]MSR-Inria Joint Centre

## Abstract

We introduce the "continuized" Nesterov acceleration, a close variant of Nesterov acceleration whose variables are indexed by a continuous time parameter. The two variables continuously mix following a linear ordinary differential equation and take gradient steps at random times. This continuized variant benefits from the best of the continuous and the discrete frameworks: as a continuous process, one can use differential calculus to analyze convergence and obtain analytical expressions for the parameters; and a discretization of the continuized process can be computed exactly with convergence rates similar to those of Nesterov original acceleration. We show that the discretization has the same structure as Nesterov acceleration, but with random parameters. We provide continuized Nesterov acceleration under deterministic as well as stochastic gradients, with either additive or multiplicative noise. Finally, using our continuized framework and expressing the gossip averaging problem as the stochastic minimization of a certain energy function, we provide the first rigorous acceleration of asynchronous gossip algorithms.

## 1 Introduction

In the last decades, the emergence of numerous applications in statistics, machine learning and signal processing has led to a renewed interest in first-order optimization methods [10]. They enjoy a low iteration cost necessary to the analysis of large datasets. The performance of first-order methods was largely improved thanks to acceleration techniques (see the review by d'Aspremont et al. [14] and the many references therein), starting with the seminal work of Nesterov [42].

Let $f : \mathbb{R}^d \to \mathbb{R}$ be a convex and differentiable function, minimized at $x_* \in \mathbb{R}^d$. We assume throughout the paper that $f$ is $L$-smooth, i.e.,

$$\forall x, y \in \mathbb{R}^d, \qquad f(y) \leqslant f(x) + \langle \nabla f(x), y - x \rangle + \frac{L}{2} \|y - x\|^2. \tag{1}$$

35th Conference on Neural Information Processing Systems (NeurIPS 2021).

In addition, we sometimes assume that $f$ is $\mu$-strongly convex for some $\mu > 0$, i.e.,

$$\forall x, y \in \mathbb{R}^d, \qquad f(y) \geqslant f(x) + \langle \nabla f(x), y - x \rangle + \frac{\mu}{2} \|y - x\|^2 . \qquad (2)$$

For the problem of minimizing $f$, gradient descent is well-known to achieve a rate $f(x_k) - f(x_*) = O(k^{-1})$ in the smooth case, and a rate $f(x_k) - f(x_*) = O((1 - \mu/L)^k)$ in the smooth and strongly convex case. In both cases, Nesterov introduced an alternative method with essentially the same iteration cost, while achieving faster rates: it converges with rate $O(k^{-2})$ in the smooth convex case and with rate $O((1 - \sqrt{\mu/L})^k)$ in the smooth and strongly convex case [43]. These rates are then optimal among all black-box first-order methods that access gradients and linearly combine them [43, 41].

Nesterov acceleration relies on several sequences of iterates—two or three, depending on the formulation—and on a clever blend of gradient steps and mixing steps between the sequences. Different interpretations and motivations underlying the precise structure of accelerated schemes were approached in many works, including [12, 24, 3, 32, 2]. A large number of these works studied continuous time equivalents of Nesterov acceleration, obtained by taking the limit when stepsizes vanish, or from a variational framework. The continuous time index $t$ of the limit allowed to use differential calculus to study the convergence of these equivalents. Examples of studies relying on continuous time interpretation include [50, 33, 54, 53, 9, 18, 48, 49, 4, 5, 57, 40].

**Continuized Nesterov acceleration.**   In this paper, we propose another continuous time equivalent to Nesterov acceleration, which we refer to as the *continuized* Nesterov acceleration, which avoids vanishing stepsizes. It is built by considering two sequences $x_t, z_t \in \mathbb{R}^d$, $t \in \mathbb{R}_{\geqslant 0}$, that continuously mix following a linear ordinary differential equation (ODE), and that take gradient steps at random times $T_1, T_2, T_3, \dots$. Thus, in this modeling, mixing and gradient steps alternate randomly.

Thanks to the continuous index $t$ and some stochastic calculus, one can differentiate averaged quantities (expectations) with respect to $t$. In particular, this leads to simple analytical expressions for the optimal parameters as functions of $t$, while the optimal parameters of Nesterov accelerations are defined by recurrence relations that are complicated to solve.

The discretization $\tilde{x}_k = x_{T_k}, \tilde{z}_k = z_{T_k}, k \in \mathbb{N}$, of the continuized process can be computed directly and exactly: the result is a recursion of the same form as Nesterov iteration, but with randomized parameters, and performs similarly to Nesterov original deterministic version both in theory and in simulations.

The continuized framework can be adapted to various settings and extensions of Nesterov acceleration. In what follows, we study how the continuized acceleration behaves in the presence of *additive* and *multiplicative* noise in the gradients. In the multiplicative noise setting, our acceleration satisfies a convergence rate similar to that of [30] and depends on the *statistical condition number* of the problem at hand. The two acceleration schemes are not directly comparable as we work in a continuized setting and only deal with pure multiplicative noise. Our analysis is nevertheless much simpler, as it closely mimics that of Nesterov acceleration.

**Application to accelerated gossip algorithms.**   The continuized modeling is natural in asynchronous parallel computing where gradient steps arrive at random times. More importantly, there are situations where the continuized version of Nesterov acceleration can be practically implemented while the original acceleration can not. In distributed settings, for instance, the total number $k$ of gradient steps taken in the network is typically not known to each particular node; an advantage of the continuized acceleration is that it requires to know only the time $t$ and not $k$.

Gossip algorithms typically feature such asynchronous and distributed behaviors [11]. In gossip problems, nodes of a network aim at computing the global average of all their values by communicating only locally (with their neighbors), and without centralized coordination. In this set-up, pairs of adjacent nodes communicate at random times, asynchronously, and in parallel, so that the total number of past communications in the network at a given time is unknown to all nodes. In this paper, we formulate the gossip problem as a stochastic optimization problem. Thanks to the continuized formalism, we naturally obtain accelerated gossip algorithms that can be implemented in an asynchronous and distributed fashion.

Synchronous gossip algorithms rely on all nodes to communicate simultaneously [19]. Accelerating synchronous gossip algorithms have been studied in previous works, including *SSDA* [47], Chebyshev

acceleration [39], Jacobi-Polynomial acceleration [7]. To that day, acceleration in the asynchronous setting has also been studied in a few works (see for instance geographic gossip [20], shift registers [37], *ESDAC* [25], and randomized Kaczmarz methods [38]). However, no algorithm in an asynchronous framework has been rigorously proven to achieve an accelerated rate for general graphs [21]. Other acceleration schemes [25, 38] relied on additional synchronizations between nodes, such as the knowledge of a global iteration counter. This departs from purely asynchronous operations, hence causing practical limitation. Our accelerated randomized gossip algorithm (Section 6) recovers the same accelerated rates, and only requires the knowledge of a common continuous-time $t \in \mathbb{R}_{\geqslant 0}$.

In this context, the continuized acceleration should be seen as a close approximation to Nesterov acceleration, that features both an insightful and convenient expression as a continuous time process and a direct implementation as a discrete iteration. We thus hope to contribute to the understanding of Nesterov acceleration. In practice, the continuized framework is relevant for handling asynchrony in decentralized optimization, where agents of a network can not share a global iteration counter, preventing accelerated decentralized and asynchronous methods.

**Notations.** The index $k$ always denotes a non-negative integer, while indices $t, s$ always denote non-negative reals.

**Structure of the paper.** In Section 2, we recall standard results on gradient descent and Nesterov acceleration. In Section 3, we introduce a continuized variant of Nesterov acceleration. In Section 4, we show that discretizing the continuized acceleration yields an iterative method similar to that of Nesterov but with random parameters. In Section 5, we study continuized Nesterov acceleration under pure-multiplicative noise. We finally present accelerated asynchronous algorithms for the gossip problem in Section 6, as well as for decentralized optimization in Section 7.

## 2 Reminders on Nesterov acceleration

For the sake of comparison, let us first recall the classical Nesterov acceleration. To improve the convergence rate of gradient descent, Nesterov introduced iterations of three sequences, parametrized by $\tau_k, \tau_k', \gamma_k, \gamma_k', k \geqslant 0$, of the form

$$y_k = x_k + \tau_k(z_k - x_k)\,, \tag{3}$$
$$x_{k+1} = y_k - \gamma_k \nabla f(y_k)\,, \tag{4}$$
$$z_{k+1} = z_k + \tau_k'(y_k - z_k) - \gamma_k' \nabla f(y_k)\,. \tag{5}$$

Depending on whether the function $f$ is known to be (1) convex, or (2) strongly convex with a known strong convexity parameter, Nesterov provided a set of parameter choices for achieving acceleration.

**Theorem 1** (Convergence of accelerated gradient descent). *Nesterov accelerated scheme satisfies:*

1. *Choose the parameters $\tau_k = 1 - \frac{A_k}{A_{k+1}}, \tau_k' = 0, \gamma_k = \frac{1}{L}, \gamma_k' = \frac{A_{k+1} - A_k}{L}, k \geqslant 0$, where the sequence $A_k, k \geqslant 0$, is defined by the recurrence relation*

$$A_0 = 0\,, \qquad\qquad A_{k+1} = A_k + \frac{1}{2}(1 + \sqrt{4A_k + 1})\,.$$

   *Then*

$$f(x_k) - f(x_*) \leqslant \frac{2L\|x_0 - x_*\|^2}{k^2}\,.$$

2. *Assume further that $f$ is $\mu$-strongly convex, $\mu > 0$. Choose the constant parameters $\tau_k \equiv \frac{\sqrt{\mu/L}}{1 + \sqrt{\mu/L}}, \tau_k' \equiv \sqrt{\frac{\mu}{L}}, \gamma_k \equiv \frac{1}{L}, \gamma_k' \equiv \frac{1}{\sqrt{\mu L}}, k \geqslant 0$. Then*

$$f(x_k) - f(x_*) \leqslant \left( f(x_0) - f(x_*) + \frac{\mu}{2}\|z_0 - x_*\|^2 \right) \left( 1 - \sqrt{\frac{\mu}{L}} \right)^k\,.$$

This result can be found as is in d'Aspremont et al. [14, Sections 4.4.1 and 4.5.3]. In the convex case, Nesterov acceleration achieves the rate $O(1/k^2)$, whereas gradient descent achieves a rate $O(1/k)$

(see [43, Corollary 2.1.2] for instance). In the strongly convex case, Nesterov acceleration achieves the rate $O((1 - \sqrt{\mu/L})^k)$, where gradient descent achieves a rate $O((1 - \mu/L)^k)$ (see [43, Theorem 2.1.15] for instance). In both cases, this results in a significant speedup in practice, see Figure 1.

From a high-level perspective, Nesterov acceleration iterates over several variables, alternating between gradient steps (always with respect to the gradient at $y_k$) and mixing steps, where the running value of a variable is replaced by a linear combination of the other variables. However, the precise way gradient and mixing steps are coupled is rather mysterious, and the success of the proof of Theorem 1 relies heavily on the detailed structure of the iterations. In the next section, we try to gain perspective on this structure by developing a continuized version of the acceleration.

## 3   Continuized version of Nesterov acceleration

This paper uses several mathematical notions related to random processes. The following sections expose the results from heuristic considerations of those notions, rigorously defined in Appendix C.

We argue that the accelerated iteration becomes more natural when considering two variables $x_t, z_t$ indexed by a continuous time $t \geqslant 0$, that are continuously mixing and that take gradient steps at random times. More precisely, let $T_1, T_2, T_3, \ldots \geqslant 0$ be random times such that $T_1, T_2 - T_1, T_3 - T_2, \ldots$ are independent identically distributed (i.i.d.), of law exponential with rate 1 (any constant rate would do, we choose 1 to make the comparison with discrete time $k$ straightforward). By convention, we choose that our stochastic processes $t \mapsto x_t, t \mapsto z_t$ are càdlàg almost surely, i.e., right continuous with well-defined left-limits $x_{t-}, z_{t-}$ (Definition 6 in Appendix C). Our dynamics are parametrized by functions $\gamma_t, \gamma'_t, \tau_t, \tau'_t, t \geqslant 0$. At random times $T_1, T_2, \ldots$, our sequences take gradient steps

$$x_{T_k} = x_{T_k-} - \gamma_{T_k} \nabla f(x_{T_k-}),\tag{6}$$

$$z_{T_k} = z_{T_k-} - \gamma'_{T_k} \nabla f(x_{T_k-}).\tag{7}$$

Because of the memoryless property of the exponential distribution, in a infinitesimal time interval $[t, t + \mathrm{d}t]$, the variables take gradients steps with probability $\mathrm{d}t$, independently of the past. Between these random times, the variables mix through a linear, translation-invariant, ordinary differential equation (ODE)

$$\mathrm{d}x_t = \eta_t(z_t - x_t)\mathrm{d}t,\tag{8}$$

$$\mathrm{d}z_t = \eta'_t(x_t - z_t)\mathrm{d}t.\tag{9}$$

Following the notation of stochastic calculus, we can write the process more compactly in terms of the Poisson point measure $\mathrm{d}N(t) = \sum_{k \geqslant 1} \delta_{T_k}(\mathrm{d}t)$, which has intensity the Lebesgue measure $\mathrm{d}t$,

$$\mathrm{d}x_t = \eta_t(z_t - x_t)\mathrm{d}t - \gamma_t \nabla f(x_t)\mathrm{d}N(t),\tag{10}$$

$$\mathrm{d}z_t = \eta'_t(x_t - z_t)\mathrm{d}t - \gamma'_t \nabla f(x_t)\mathrm{d}N(t).\tag{11}$$

Before giving convergence guarantees for such processes, let us digress quickly on why we can expect an iteration of this form to be mathematically appealing.

First, from a Markov chain indexed by a discrete time index $k$, one can associate the so-called *continuized* Markov chain, indexed by a continuous time $t$, that makes transition with the same Markov kernel, but at random times, with independent exponential time intervals [1]. Following this terminology, we refer to our acceleration (10)-(11) as the continuized acceleration. The continuized Markov chain is appreciated for its continuous time parameter $t$, while keeping many properties of the original Markov chain; similarly the continuized acceleration is arguably simpler to analyze, while performing similarly to Nesterov acceleration.

Second, it can also be compared with coordinate gradient descent methods, that are easier to analyze when coordinates are selected randomly rather than in an ordered way [55]. Similarly, the continuized acceleration is simpler to analyze because the gradient steps (6)-(7) and the mixing steps (8)-(9) alternate randomly, due to the randomness of $T_k, k \geqslant 0$.

In analogy with Theorem 1, we give choices of parameters that lead to accelerated convergence rates, in the convex case (1) and in the strongly convex case (2). Convergence is analyzed as a function of $t$. As $\mathrm{d}N(t)$ is a Poisson point process with rate 1, $t$ is the expected number of gradient steps done by the algorithm. Thus $t$ is analogous to $k$ in Theorem 1. In the theorem below, $\mathbb{E}$ denotes the expectation with respect to the Poisson point process $\mathrm{d}N(t)$, the only source of randomness.

**Theorem 2** (Convergence of continuized Nesterov acceleration). *The continuized Nesterov acceleration satisfies the following two points.*

1. *Choose the parameters $\eta_t = \frac{2}{t}, \eta'_t = 0, \gamma_t = \frac{1}{L}, \gamma'_t = \frac{t}{2L}$. Then*

$$\mathbb{E}f(x_t) - f(x_*) \leqslant \frac{2L\|z_0 - x_*\|^2}{t^2} \, .$$

2. *Assume further that $f$ is $\mu$-strongly convex, $\mu > 0$. Choose the constant parameters $\eta_t = \eta'_t \equiv \sqrt{\frac{\mu}{L}}, \gamma_t \equiv \frac{1}{L}, \gamma'_t \equiv \frac{1}{\sqrt{\mu L}}$. Then*

$$\mathbb{E}f(x_t) - f(x_*) \leqslant \left( f(x_0) - f(x_*) + \frac{\mu}{2}\|z_0 - x_*\|^2 \right) \exp\left( -\sqrt{\frac{\mu}{L}}t \right) \, .$$

We give an elementary sketch of proof in Appendix D.1 and a complete proof in Appendix D.2. Many authors have proposed continuous-time versions of Nesterov acceleration using differential calculus, see the numerous references in the introduction. For instance, in Su et al. [50], an ODE is obtained from Nesterov acceleration by taking the joint asymptotic where the stepsizes vanish and the number of iterates is rescaled. The resulting ODE must be discretized to be implemented; choosing the right discretization is not straightforward as it introduces stability and approximation errors that must be controlled [57, 49, 46].

On the contrary, our continuous time process (10)-(11) does not correspond to a limit where the stepsizes vanish. However, in Appendix F, we check that the random continuized acceleration has the same deterministic ODE scaling limit as Nesterov acceleration. This sanity check emphasizes that the continuized acceleration is fundamentally different from previous continuous-time equivalents.

**Remark 1.** *A similar Markovian structure can be obtained in a discrete setting by flipping i.i.d. coins to trigger gradient steps. By denoting $p > 0$ the probability to trigger a gradient step when flipping a coin,* (i) *$p = 1$ gives the classical setting, and* (ii) *$p \to 0$ while renormalizing time gives our continuized framework. In fact, this setting with updates triggered randomly is an interpolation between the classical and continuized frameworks, and consists in replacing exponential random variables by geometric random variables of parameter $p$ for the waiting-time between updates. We thus believe the convergence guarantees described here and in the following can be adapted for this discrete scheme.*

## 4  Discrete implementation of the continuized acceleration with random parameters

In this section, we show that the continuized acceleration can be implemented exactly as a discrete algorithm. This contrasts with the discretization of ODEs that introduces discretization errors; here, we compute exactly

$$\tilde{x}_k := x_{T_k}, \qquad\qquad \tilde{y}_k := x_{T_{k+1}-}, \qquad\qquad \tilde{z}_k := z_{T_k},$$

with the convention that $T_0 = 0$. The three sequences $\tilde{x}_k, \tilde{y}_k, \tilde{z}_k, k \geqslant 0$, satisfy a recurrence relation of the same structure as Nesterov acceleration, but with random weights. The resulting randomized discrete algorithm satisfies performance guarantees similar to those of Nesterov acceleration.

**Theorem 3** (Discrete version of continuized acceleration). *For any stochastic process of the form (10)-(11), we have*

$$\tilde{y}_k = \tilde{x}_k + \tau_k(\tilde{z}_k - \tilde{x}_k), \tag{12}$$

$$\tilde{x}_{k+1} = \tilde{y}_k - \tilde{\gamma}_k \nabla f(\tilde{y}_k), \tag{13}$$

$$\tilde{z}_{k+1} = \tilde{z}_k + \tau'_k(\tilde{y}_k - \tilde{z}_k) - \tilde{\gamma}'_k \nabla f(\tilde{y}_k), \tag{14}$$

*for some random parameters $\tau_k, \tau'_k, \tilde{\gamma}_k, \tilde{\gamma}'_k$ (that are functions of $T_k, T_{k+1}, \eta_t, \eta'_t, \gamma_t, \gamma'_t$).*

1. *For the parameters of Theorem 2.(1), $\tau_k = 1 - \left(\frac{T_k}{T_{k+1}}\right)^2, \tau'_k = 0, \tilde{\gamma}_k = \frac{1}{L}$, and $\tilde{\gamma}'_k = \frac{T_k}{2L}$. Then*

$$\mathbb{E}\left[ T_k^2 \left( f(\tilde{x}_k) - f(x_*) \right) \right] \leqslant 2L\|z_0 - x_*\|^2 \, .$$

2. *For the parameters of Theorem 2.(2),* $\tau_k = \frac{1}{2}\left(1 - \exp\left(-2\sqrt{\frac{\mu}{L}}(T_{k+1} - T_k)\right)\right),$
   $\tau'_k = \tanh\left(\sqrt{\frac{\mu}{L}}(T_{k+1} - T_k)\right), \tilde{\gamma}_k = \frac{1}{L},$ *and* $\tilde{\gamma}'_k = \frac{1}{\sqrt{\mu L}}.$ *Then*

$$\mathbb{E}\left[\exp\left(\sqrt{\frac{\mu}{L}}T_k\right)(f(\tilde{x}_k) - f(x_*))\right] \leqslant f(x_0) - f(x_*) + \frac{\mu}{2}\|z_0 - x_*\|^2.$$

The law of $T_k$ is well known: it is the sum of $k$ i.i.d. random variables of law exponential with rate 1; this is called an Erlang or Gamma distribution with shape parameter $k$ and rate 1. One can use well-known properties of this law, such as its concentration around its expectation $\mathbb{E}T_k = k$, to derive corollaries of the bounds above. The performance guarantees are proved in Appendix D.2, and the formula for the discretization is studied in E. In Appendix A.1, we provide simulations confirming that this discrete random algorithm has a performance similar to Nesterov's original acceleration.

## 5 Continuized Nesterov acceleration of stochastic gradient descent

We now investigate the design of continuized accelerations of stochastic gradient descent. We assume that we do not have direct access to the gradient $\nabla f(x)$ but to a random estimate $\nabla f(x, \xi)$, where $\xi \in \Xi$ is random of law $\mathcal{P}$. In the continuized framework, the randomness of the stochastic gradient and its time mix in a particularly convenient way. For similar reasons, Latz studied stochastic gradient descent as a gradient flow on a random function that is regenerated at a Poisson rate [35]. However, this approach has the same shortcomings as the other approaches based on gradient flows: the subsequent discretization introduces non-trivial errors. We avoid this problem here.

We keep the algorithms of the same form, replacing gradients by stochastic gradients. Let $\xi_1, \xi_2, \ldots$ be i.i.d. random variables of law $\mathcal{P}$. We take stochastic gradient steps at the random times $T_1, T_2, \ldots,$

$$x_{T_k} = x_{T_k-} - \gamma_{T_k}\nabla f(x_{T_k-}, \xi_k),$$
$$z_{T_k} = z_{T_k-} - \gamma'_{T_k}\nabla f(x_{T_k-}, \xi_k).$$

Between these random times, the variables mix through the same ODE

$$\mathrm{d}x_t = \eta_t(z_t - x_t)\mathrm{d}t,$$
$$\mathrm{d}z_t = \eta'_t(x_t - z_t)\mathrm{d}t.$$

This can be written more compactly in terms of the Poisson point measure $\mathrm{d}N(t, \xi) = \sum_{k \geqslant 1}\delta_{(T_k, \xi_k)}(\mathrm{d}t, \mathrm{d}\xi)$ on $\mathbb{R}_{\geqslant 0} \times \Xi$, which has intensity $\mathrm{d}t \otimes \mathcal{P},$

$$\mathrm{d}x_t = \eta_t(z_t - x_t)\mathrm{d}t - \gamma_t\int_{\Xi}\nabla f(x_t, \xi)\mathrm{d}N(t, \xi), \tag{15}$$

$$\mathrm{d}z_t = \eta'_t(x_t - z_t)\mathrm{d}t - \gamma'_t\int_{\Xi}\nabla f(x_t, \xi)\mathrm{d}N(t, \xi). \tag{16}$$

Here, the discussion depends on the properties satisfied by the stochastic gradients $\nabla f(x, \xi)$. In Appendix B, we study the so-called *additive noise* case. We show that the continuized acceleration satisfies perturbed convergence rates with the same choices of parameters as in Theorem 2. We thus show some robustness of the above acceleration to additive noise. Instead, in this section, we focus on the so-called *pure multiplicative noise* case, as it is crucial for the study of asynchronous gossip that follows. In this setting, parameters need to be chosen differently for our proof technique to work. A continuized acceleration is still possible, depending on the statistical condition number.

We now focus on functions $f$ is of the following form, typical to least-squares supervised learning:

$$\forall x \in \mathbb{R}^d, \ f(x) = \mathbb{E}_{(a,b)\sim\mathcal{P}}\left[\frac{1}{2}(b - \langle x, a\rangle)^2\right], \tag{17}$$

where $\xi = (a, b) \in \mathbb{R}^d \times \mathbb{R}$ is random of law $\mathcal{P}$. We assume that our *stochastic first order oracle* is the gradient of one realization of the expectation, namely,

$$\nabla f(x, \xi) = -(b - \langle x, a\rangle)a, \qquad \xi = (a, b).$$

We investigate *noiseless*—or purely multiplicative—stochastic gradients, in the sense that almost surely, for $\xi = (a, b) \sim \mathcal{P}$:

$$b = \langle x_*, a\rangle, \text{ so that } \nabla f(x_*, \xi) = 0. \tag{18}$$

Noiseless stochastic gradients are relevant in several situations, such as coordinate gradient descent with randomly sampled coordinates [51, 44, 55] (where $\nabla f(x, \xi) = m\langle \nabla f(x), e_i \rangle e_i$ with $i$ uniformly random in $\{1, \ldots, d\}$), over-parameterized regime for least squares regression [52], function interpolation and gossip algorithms [8].

For a symmetric non-negative matrix $A$ and a vector $x$, we denote $\|x\|_A^2 = x^\top A x$. Let $H = \mathbb{E}[aa^\top]$ be the Hessian of $f$. Let $R^2$ be the smallest positive real number such that:

$$\mathbb{E}\left[\|a\|^2 aa^\top\right] \preccurlyeq R^2 H. \tag{19}$$

Further, similarly to Jain et al. [30], we define the statistical condition number of the problem as the smallest $\tilde{\kappa} > 0$ such that:

$$\mathbb{E}\left[\|a\|_{H^{-1}}^2 aa^\top\right] \preccurlyeq \tilde{\kappa} H. \tag{20}$$

**Theorem 4** (Continuized acceleration with pure multiplicative noise). *Assume that* (18), (19) *and* (20) *hold true. Then the continuized acceleration satisfies the following.*

1. *Choose the parameters* $\eta_t = \frac{2}{t}, \eta_t' = 0, \gamma_t = \frac{1}{R^2}, \gamma_t' = \frac{t}{2R^2\tilde{\kappa}}$. *Then*

$$\frac{1}{2}\mathbb{E}\|x_t - x_*\|^2 \leqslant \frac{R^2\tilde{\kappa}\|z_0 - x_*\|_{H^{-1}}^2}{t^2}.$$

2. *Assume further that $f$ is $\mu$-strongly convex, i.e., all eigenvalues of $H$ are greater or equal to $\mu$, where $\mu > 0$. The condition number of $f$ is then defined as $\kappa = R^2/\mu$. For the parameters $\eta_t = \eta_t' = \frac{1}{\sqrt{\kappa\tilde{\kappa}}}, \gamma_t = \frac{1}{R^2}$ and $\gamma_t' = \frac{1}{R^2}\sqrt{\frac{\kappa}{\tilde{\kappa}}}$, we have:*

$$\frac{1}{2}\mathbb{E}\|x_t - x_*\|^2 \leqslant \left(\frac{1}{2}\|x_0 - x_*\|^2 + \frac{\mu}{2}\|z_0 - x_*\|_{H^{-1}}^2\right)\exp\left(-\frac{t}{\sqrt{\kappa\tilde{\kappa}}}\right).$$

In the strongly convex case, the benefits of this acceleration are similar to those of Jain et al. [30] with classical discrete iterates: while stochastic gradient descent with stepsize $1/R^2$ is easily shown to achieve an exponential rate of convergence $1/\kappa$, the acceleration enjoys a rate of convergence of $1/\sqrt{\kappa\tilde{\kappa}}$. Note that from the definitions, $\tilde{\kappa} \leqslant \kappa$, thus the acceleration performs as least as well as the naive algorithm. However, depending on the distribution of $a$, the improvement might either be significant or null. We refer the reader to the rich discussion in Jain et al. [30] which provides insights on the interpretation of $\tilde{\kappa}$ and on the possibility to accelerate. Below, we provide a complementary perspective on the statistical condition number in the context of gossip algorithms, where it can be interpreted in terms of effective resistances of graphs.

Albeit more restrictive in terms of assumptions, our analysis is much simpler than that of Jain et al. [30], as it relies on a standard Lyapunov function, similar to that of the continuized acceleration (Theorem 2). In Appendix G, we use the same analysis framework to prove convergence of accelerated coordinate descent, which is another noiseless stochastic method.

## 6  Accelerating Randomized Gossip

The continuized framework allows designing accelerated decentralized algorithms requiring synchronized clocks, but no synchronization of the communications. In this section, we illustrate this statement in the simple case of gossip algorithms; the more general case of decentralized optimization is discussed in the next section.

Let $G = (\mathcal{V}, \mathcal{E})$ a connected graph representing a communication network of agents. Each agent $v \in \mathcal{V}$ is assigned a real number $x_0(v) \in \mathbb{R}$. The goal of the averaging (or gossip) problem is to design an iterative procedure allowing each agent of the network to know the average $\bar{x} = \frac{1}{m}\sum_{v \in \mathcal{V}} x_0(v)$ using only local communications, *i.e.*, communications between adjacent agents in the network.

We formalize the communication model of randomized gossip [11]. Time $t$ is indexed continuously in $\mathbb{R}_{\geqslant 0}$. We generate a Poisson point measure $dN(t, e) = \sum_{k \geqslant 1} \delta_{(T_k, \{v_k, w_k\})}$ with intensity measure $dt \otimes \mathcal{P}$, where $dt$ is the Lebesgue measure on $\mathbb{R}_{\geqslant 0}$ and $\mathcal{P} = (\mathcal{P}_{\{v,w\}})_{\{v,w\} \in \mathcal{E}}$ is a probability measure on the set $\mathcal{E}$ of edges. For $k \geqslant 0$, $T_k$ is a time at which edge $\{v_k, w_k\}$ is *activated*: adjacent nodes $v_k$

and $w_k$ can communicate and perform a pairwise update. The Poisson point measure assumption implies that edges are activated independently of one another and from the past: the activation times of edge $\{v, w\}$ form a Poisson point process of intensity $\mathcal{P}_{\{v,w\}}$.

To solve the gossip problem, Boyd et al. [11] proposed the following naive strategy: each agent $v \in \mathcal{V}$ keeps a local estimate $x_t(v)$ of the average and, upon activation of edge $\{v_k, w_k\}$ at time $T_k \in \mathbb{R}_{\geqslant 0}$, the activated nodes $v_k, w_k$ average their current estimates

$$x_{T_k}(v_k), \, x_{T_k}(w_k) \quad \longleftarrow \quad \frac{x_{T_k-}(v_k) + x_{T_k-}(w_k)}{2}\,.$$

In this section, we accelerate this naive procedure. Our strategy is to apply Section 5 as follows. Consider the energy function

$$f(x) = \sum_{\{v,w\}\in\mathcal{E}} \frac{\mathcal{P}_{\{v,w\}}}{2}(x(v) - x(w))^2\,, \qquad x = (x(v))_{v\in\mathcal{V}}\,. \tag{21}$$

This function is convex, smooth, and writes in the form (17):

$$f(x) = \mathbb{E}_{\{v,w\}\sim\mathcal{P}}\left[\frac{1}{2}\left\langle x, a_{\{v,w\}}\right\rangle^2\right], \tag{22}$$

where $a_{\{v,w\}} = e_v - e_w$ and $(e_v)_{v\in\mathcal{V}}$ forms the canonical basis of $\mathbb{R}^{\mathcal{V}}$. As in Section 5, a stochastic gradient of $f$ is obtained by taking the gradient of one realization of the expectation, namely:

$$\nabla f(x, \{v,w\}) = \langle x, a_{\{v,w\}}\rangle a_{\{v,w\}} = \begin{cases} x(v) - x(w) & \text{at coordinate } v, \\ x(w) - x(v) & \text{at coordinate } w, \\ 0 & \text{at all other coordinates.} \end{cases} \tag{23}$$

As a consequence, a stochastic gradient step with stepsize $1/2$ corresponds to a local averaging alongside edge $\{v, w\}$, where $\{v, w\} \sim \mathcal{P}$. More generally, the randomized gossip algorithm as described by Boyd et al. [11] is the stochastic gradient descent:

$$\mathrm{d}x_t = -\frac{1}{2}\int_{\mathbb{R}_{\geqslant 0}\times\mathcal{E}} \nabla f(x_t, \{v,w\})\mathrm{d}N(t, \{v,w\})\,. \tag{24}$$

Using Section 5, we can accelerate this algorithm if we know the strong convexity parameter of $f$ and the constants $R^2$ and $\tilde{\kappa}$ as defined in (19) and (20) respectively. These constants can be intepreted as graph-related quantities here.

**Definition 1** (Graph-related quantities). *The Laplacian matrix $\mathcal{L} \in \mathbb{R}^{\mathcal{V}\times\mathcal{V}}$ of graph $G$ with weights $(\mathcal{P}_{\{v,w\}})_{\{v,w\}\in\mathcal{E}}$ on the edges is the matrix with entries $\mathcal{L}_{v,w} = -\mathcal{P}_{\{v,w\}}$ if $\{v, w\} \in \mathcal{E}$, $\mathcal{L}_{v,v} = \sum_{w\sim v} \mathcal{P}_{\{v,w\}}$, and $\mathcal{L}_{v,w} = 0$ if $\{v, w\} \notin \mathcal{E}$. We denote $\mu_{\text{gossip}}$ the second smallest eigenvalue of $\mathcal{L}$, corresponding to its smallest positive eigenvalue. For $\{v, w\} \in \mathcal{E}$, let $R_{\text{eff}}(v, w) = (e(v) - e(w))^\top \mathcal{L}^{-1}(e(v) - e(w))$ be the effective resistance of edge $\{v, w\}$, and $R_{\max} = \max_{\{v,w\}\in\mathcal{E}} R_{\text{eff}}(v, w)$ be the maximal resistance in the graph.*

The function $f$ is quadratic with Hessian $\mathcal{L}$, and strongly convex with parameter $\mu_{\text{gossip}}$ on the hyperplane $F = \{x \in \mathbb{R}^{\mathcal{V}} : \sum_{v\in\mathcal{V}} x(v) = \bar{x}\}$; hence we use the (perhaps abusive) notation $\mu_{\text{gossip}}$ throughout. Moreover, the conditions (19) and (20) are satisfied with $R^2 = 2$, $\tilde{\kappa} = R_{\max}$.

These parameters being given, the accelerated stochastic gradient descent updates (15)-(16) can be instantiated as follows. Each agent $v \in \mathcal{V}$ keeps two local estimates $x_t(v), z_t(v)$ of $\bar{x}$, initialized at $x_0(v)$. Upon activation of edge $\{v_k, w_k\}$ at time $T_k$,

$$x_{T_k}(v_k) = x_{T_k}(w_k) = \frac{x_{T_k-}(v_k) + x_{T_k-}(w_k)}{2}\,,$$

$$z_{T_k}(v_k) = z_{T_k-}(v_k) + \frac{1}{\sqrt{2\mu_{\text{gossip}}R_{\max}}}(x_{T_k-}(w_k) - x_{T_k-}(v_k))\,,$$

$$z_{T_k}(w_k) = z_{T_k-}(w_k) + \frac{1}{\sqrt{2\mu_{\text{gossip}}R_{\max}}}(x_{T_k-}(v_k) - x_{T_k-}(w_k))\,.$$

Between these updates, $x_t(v)$ and $z_t(v)$ locally mix at all nodes $v \in \mathcal{V}$, according to the coupled ODE:

$$\mathrm{d}x_t(v) = \sqrt{\frac{2\mu_{\text{gossip}}}{R_{\max}}}(z_t(v) - x_t(v))\mathrm{d}t,$$

$$\mathrm{d}z_t(v) = \sqrt{\frac{2\mu_{\text{gossip}}}{R_{\max}}}(x_t(v) - z_t(v))\mathrm{d}t.$$

This algorithm is *asynchronous* in the sense that it does not require global synchronous operations: the mixing of local variables does not require any synchronization since parameter $t \in \mathbb{R}_{\geqslant 0}$ is available at all nodes independently from the number of past updates, while a local pairwise update between adjacent nodes $v$ and $w$ only requires a local synchronization.

**Theorem 5** (Accelerated randomized gossip). *Let $(x_t(v))_{v \in \mathcal{V}, t \geqslant 0}$ be generated with accelerated randomized gossip. For any $t \in \mathbb{R}_{\geqslant 0}$:*

$$\sum_{v \in \mathcal{V}} \frac{1}{2}\mathbb{E}\left[(x_t(v) - \bar{x})^2\right] \leqslant 2\left(\sum_{v \in \mathcal{V}} \frac{1}{2}(x_0(v) - \bar{x})^2\right)\exp\left(-\sqrt{\frac{\mu_{\text{gossip}}}{2R_{\max}}}t\right).$$

Let $\theta_{\text{ARG}} = \sqrt{\frac{\mu_{\text{gossip}}}{2R_{\max}}}$ be the rate of convergence of accelerated randomized gossip, and $\theta_{\text{RG}} = \mu_{\text{gossip}}$ be the rate of convergence of randomized gossip [11]. We have $\theta_{\text{ARG}} \geqslant \theta_{\text{RG}}/\sqrt{2}$. Let us exhibit scenarios over which accelerated randomized gossip gains several orders of magnitude. Denoting $\mathcal{P}_{\min} = \min_{\{v,w\} \in \mathcal{E}} \mathcal{P}_{\{v,w\}}$, Ellens et al. [22] ensures that for $\{v, w\} \in \mathcal{E}$, $\mathcal{P}_{\min} R_{\text{eff}}(v, w) \leqslant 1$, so that $R_{\max} \leqslant \mathcal{P}_{\min}^{-1}$.

**Corollary 1** (Comparison with randomized gossip). *Accelerated randomized gossip achieves a rate satisfying:*

$$\sqrt{\frac{\theta_{\text{RG}}\mathcal{P}_{\min}}{2}} \leqslant \theta_{\text{ARG}}.$$

*Assume furthermore that there exist some constants $c > 0$ such that for all $\{v, w\} \in \mathcal{E}$, $\mathcal{P}_{\{v,w\}} \leqslant c\mathcal{P}_{\min}$ and $d_v + d_w \leqslant 2d$. Then, with $C = 1/\sqrt{2cd}$:*

$$C\sqrt{\frac{\theta_{\text{RG}}}{|\mathcal{V}|}} \leqslant \theta_{\text{ARG}}.$$

Assume now for simplicity that the Poisson intensities $\mathcal{P}_{\{v,w\}}$ are all equal to $1/|\mathcal{E}|$. Denoting $|\mathcal{V}| = m$, on the cyclic and the line graph, this gives us $\theta_{\text{ARG}} = \Omega(1/m^2)$ while $\theta_{\text{RG}} \asymp 1/m^3$. On a $d$-dimensional grid, we have $\theta_{\text{ARG}} = \Omega(1/m^{1+1/d})$ and $\theta_{\text{RG}} \asymp 1/m^{1+2/d}$. However, on graphs with unbounded degrees, no improvements are observed, as illustrated in Figure 2, Appendix A.2. In the case of the complete graph, this is expected since at least $\theta_{\text{RG}}^{-1} \asymp m$ communications are needed to compute the average. We thus recover the same rates as Dimakis et al. [20] for the graphs they study, but generalized to any network.

## 7 Accelerating Asynchronous Decentralized Optimization

Our continuized framework for accelerating randomized gossip can be extended to the more general problem of decentralized optimization: each node $v$ in the network $G$ previously defined holds a function $f_v : \mathbb{R}^d \to \mathbb{R}$, $\mu$-strongly convex and $L$-smooth. Nodes of the network collaborate to solve:

$$\min_{x \in \mathbb{R}^d} \left\{ f(x) = \frac{1}{|\mathcal{V}|} \sum_{v \in \mathcal{V}} f_v(x) \right\}. \tag{25}$$

As in gossip averaging, only local communications are allowed. Quantities related to $f_v$ can only be computed at node $v$. In the case of empirical risk minimization, $f_v$ represents the empirical risk related to node $v$'s local data. Setting $f_v(x) = \frac{1}{2}\|x - x_0(v)\|^2$ leads to the averaging problem previously described. Similarly to Section 6, time is indexed continuously by $t$ in $\mathbb{R}_{\geqslant 0}$, and communications are

ruled by the same Poisson point measure $dN(t, e) = \sum_{k \geqslant 1} \delta_{(T_k, \{v_k, w_k\})}$ on $\mathbb{R}_{\geqslant 0} \times \mathcal{E}$. Yet, we no longer assume (as in Theorem 4) that the function $f$ is quadratic. Instead, we write a dual formulation of Problem (25) and minimize it using a continuized version of accelerated coordinate descent [45] that we present in Appendix G. This leads to an accelerated decentralized algorithm to solve (25). Our algorithm mimics the behavior of accelerated randomized gossip: a node possesses two local parameters that mix continuously through a time-independent ODE. At time $T_k$, adjacent nodes $v_k$ and $w_k$ use their local function in order to compute gradient conjugates $\nabla f_v^*(x(v)), \nabla f_w^*(x(w))$. Since the local functions are not simple quadratics anymore, the stochastic gradients $\nabla f(x, \{v, w\})$ from Equation (26) are replaced by terms proportional to:

$$G(y, \{v, w\}) = \begin{cases} \nabla f_v^*(y(v)) - \nabla f_w^*(y(w)) & \text{at coordinate } v, \\ -\nabla f_v^*(y(v)) - \nabla f_w^*(y(w)) & \text{at coordinate } w, \\ 0 & \text{at all other coordinates.} \end{cases} \qquad (26)$$

Due to lack of space, we describe the iterations more in details in Appendix H, together with a relevant choice of parameters. The crucial point is that, similarly to the gossip averaging case, we do not require nodes to be aware of a global iteration counter. Yet, we still obtain the same convergence rate as [25], as provided by the following theorem. The same approach can be used to "continuize" other accelerated randomized gossip algorithms for decentralized optimization, such as ADFS [26].

**Theorem 6** (Accelerated asynchronous decentralized optimization)**.** *For* $(x_t(v))_{v \in \mathcal{V}} = (\nabla f_v^*(z_t(v)))_{v \in \mathcal{V}}$ *generated by the accelerated coordinate descent on the dual of Problem* (25)*:*

$$\sum_{v \in \mathcal{V}} \frac{1}{2} \mathbb{E}\left[\|x_t(v) - x_*\|^2\right] \leqslant C \left(\sum_{v \in \mathcal{V}} \frac{1}{2} \|x_0(v) - x_*\|^2\right) \exp\left(-\frac{\theta'_{\mathrm{ARG}}}{\sqrt{\kappa}} t\right),$$

*where* $\kappa = \mu/L$ *is an upper bound on the condition number of* $f$*,* $C$ *is a constant that depends on the graph and* $\kappa$*, and* $\theta'_{\mathrm{ARG}}$ *is the rate of convergence of accelerated randomized gossip on the graph* $G$ *as defined in Theorem 5 but with graph resistances are defined in a different way (see Theorem 10).*

## 8 Conclusion

In this work, we introduced a continuized version of Nesterov's accelerated gradients. In a nutshell, the method has two sequences of iterates which take gradient steps at random times. In between gradient steps, the two sequences mix following a simple ordinary differential equation, whose parameters are picked to ensure good convergence properties of the method.

As compared to other continuous time models of Nesterov acceleration, a key feature of this approach is that the method can be implemented without any approximation, as the differential equation governing the mixing procedure has a simple analytical solution. A discretization of the continuized method corresponds to an accelerated gradient method with random parameters.

Continuization strategies were introduced in the context of Markov chains [1]. Here, they allow using acceleration mechanisms in asynchronous distributed optimization, where usually agents are not aware of the total number of iterations taken so far. This is showcased in the context of asynchronous gossip algorithms.

**Acknowledgements:** The authors thank Sam Power for pointing out the class of piecewise deterministic Markov processes and related references, and an anonymous reviewer for suggesting Remark 1. This work was funded in part by the French government under management of Agence Nationale de la Recherche as part of the "Investissements d'avenir" program, reference ANR-19-P3IA-0001(PRAIRIE 3IA Institute). We also acknowledge support from the European Research Council (grant SEQUOIA 724063), from the DGA, and from the MSR-INRIA joint centre.

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
