## A   Numerical Simulations

### A.1   Simulations of the discretized continuized acceleration

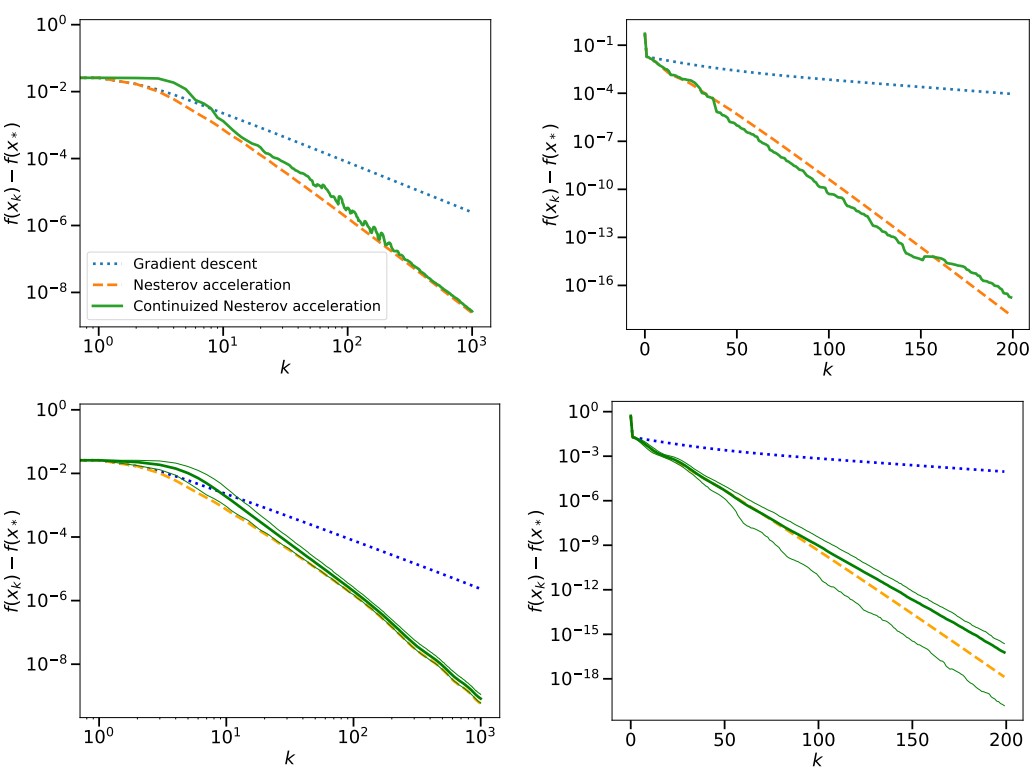

Figure 1: Comparison between gradient descent, Nesterov acceleration, and the continuized version of Nesterov acceleration, on a convex function (left plots) and a strongly convex function (right plots). For the continuized acceleration, which is randomized, the results shown in the above plots correspond to a single run. In the plots below, the thick line represents the average performance over $N = 1000$ runs of the continuized acceleration, while the thin lines represent the $5\%$ and $95\%$ quantiles.

In Figure 1, we compare this continuized Nesterov acceleration (12)-(14) with the classical Nesterov acceleration (3)-(5) and gradient descent. In the strongly convex case (right), we run the algorithms with the parameters of Theorem 1.(2) and 3.(2) on the function

$$f(x_1, x_2, x_3) = \frac{\mu}{2}(x_1 - 1)^2 + \frac{3\mu}{2}(x_2 - 1)^2 + \frac{L}{2}(x_3 - 1)^2 \,,$$

with $\mu = 10^{-2}$ and $L = 1$. In the convex case, we run the algorithms with the parameters of Theorem 1.(1) and 3.(1) on the function

$$f(x_1, \dots, x_{100}) = \frac{1}{2} \sum_{i=1}^{100} \frac{1}{i^2} \left( x_i - \frac{1}{i} \right)^2 \,,$$

which has negligible strong convexity parameter. All iterations were initialized from $x_0 = z_0 = 0$.

## A.2 Simulation of Accelerated Randomized Gossip

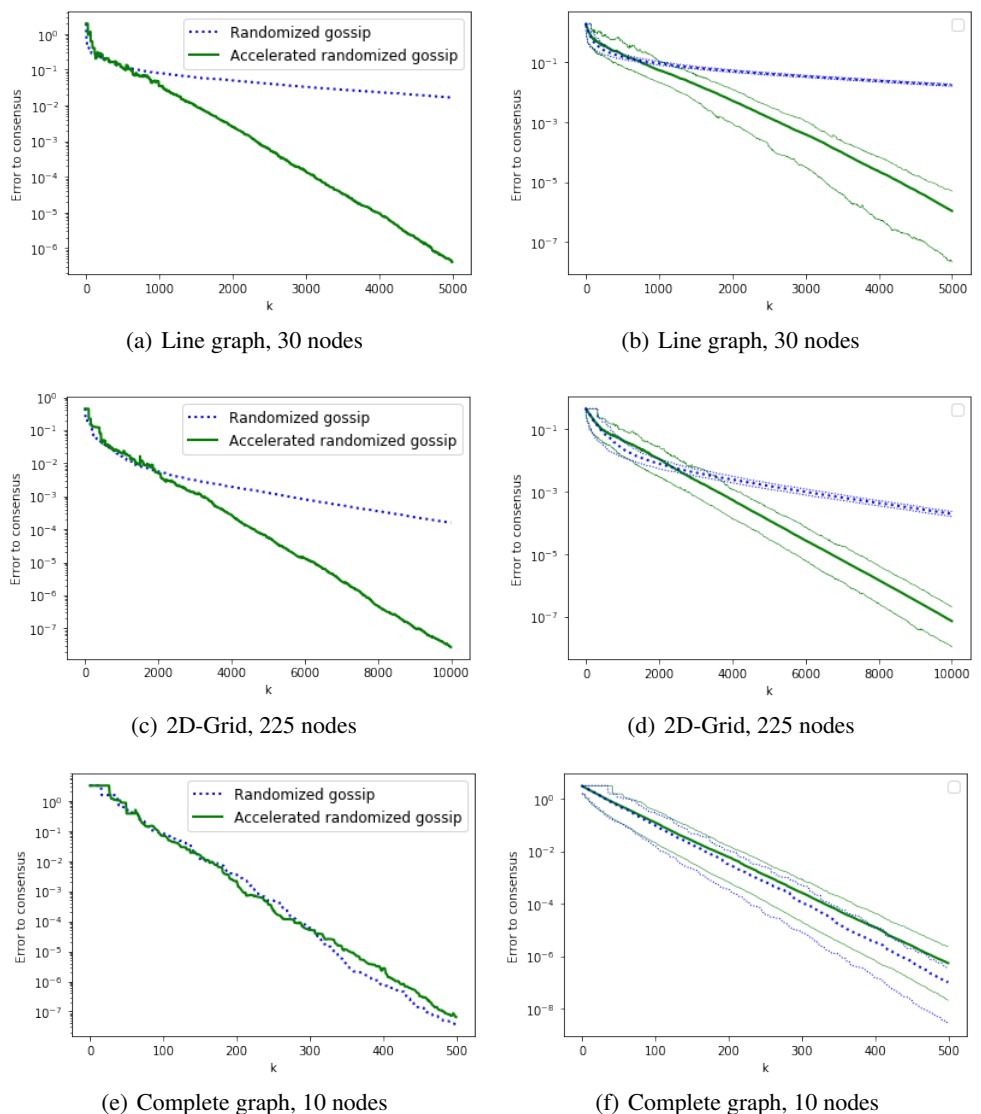

Figure 2: Comparison between randomized gossip [11] and accelerated randomized gossip from Section 6, on 3 different graphs: line with 30 nodes, 2D-Grid with 225 nodes and complete graph with 30 nodes. The probability $\mathcal{P}$ on the set of edges that determines at every activation which edge is activated is uniform in all cases. Parameters of the algorithm are taken as in Theorem 5. In all simulations, initialization was taken with a vector $x_0$ such that $x_0(v) = 0$ at all nodes, except one where $x_0(v) = 1$. Figures on the left represent one run of the algorithms. Figures on the right represent the average performance (thick line) for $N = 1000$ runes with the same settings, and the $5\%$ and $95\%$ quantiles (thin lines). As expected, we observe acceleration one the line and the grid, but no such phenomenon on the complete graph.

## B   Robustness of the continuized Nesterov acceleration to additive noise

In this section, we study the continuized acceleration (15)-(16) under stochastic gradients. We assume that our gradient estimates are unbiased, i.e.,

$$\forall x \in \mathbb{R}^d, \qquad \mathbb{E}_\xi \nabla f(x, \xi) = \nabla f(x), \tag{27}$$

and has a uniformly bounded variance, i.e., there exists $\sigma^2 \geqslant 0$ such that

$$\forall x \in \mathbb{R}^d, \qquad \mathbb{E}_\xi \left\| \nabla f(x,\xi) - \nabla f(x) \right\|^2 \leqslant \sigma^2 \, . \tag{28}$$

These assumptions typically hold in the additive noise model, where $\nabla f(x,\xi) = \nabla f(x) + \xi$, and $\xi \in \mathbb{R}^d$ satisfies $\mathbb{E}\xi = 0$, $\mathbb{E}\|\xi\|^2 \leqslant \sigma^2$. By an abuse of terminology, we say that our stochastic gradients have "additive noise" when (27) and (28) hold.

We should emphasize that similar studies of Nesterov acceleration under additive noise has been done [34, 27, 56, 17, 13, 6].

**Theorem 7** (Continuized acceleration with additive noise)**.** *Assume that the stochastic gradients are unbiased (27) and have a variance uniformly bounded by $\sigma^2$ (28). Then the continuized acceleration (15)-(16) satisfies the following.*

1. *For the parameters of Theorem 2.(1),*

$$\mathbb{E}f(x_t) - f(x_*) \leqslant \frac{2L\|z_0 - x_*\|^2}{t^2} + \sigma^2 \frac{t}{3L} \, .$$

2. *Assume further that $f$ is $\mu$-strongly convex, $\mu > 0$. For the parameters of Theorem 2.(2),*

$$\mathbb{E}f(x_t) - f(x_*) \leqslant \left( f(x_0) - f(x_*) + \frac{\mu}{2}\|z_0 - x_*\|^2 \right) \exp\left( -\sqrt{\frac{\mu}{L}}t \right) + \sigma^2 \frac{1}{\sqrt{\mu L}} \, .$$

This theorem is proved in Appendix D.3.

In the above bounds, $L$ is a parameter of the algorithm, that can be taken greater than the best known smoothness constant of the function $f$. Increasing $L$ reduces the stepsizes of the algorithm and performs some variance reduction. If the bound $\sigma^2$ on the variance is known, one can choose $L$ optimizing the above bounds in order to obtain algorithms that adapt to additive noise.

In Figure 3, we run the same simulations as in Figure 1, with two differences: (1) we add isotropic Gaussian noise on the gradients, with covariance $10^{-4}$ Id, and (2) we initialized algorithms at the optimum, i.e., $x_0 = z_0 = x_*$. Initializing at the optimum enables to isolate the effect of the additive noise only. These simulations confirm Theorem 7: the noise term is (sub-)linearly increasing in the convex case and constant in the strongly convex case.

Note that similarly to Theorem 3, one could obtain convergence bounds for the discrete implementation under the presence of additive noise.

## C   Stochastic calculus toolbox

In this appendix, we give a short introduction to the mathematical tools that we use in this paper. For more details, the reader can consult the more rigorous monographs of Jacod and Shiryaev [29], Ikeda and Watanabe [28], Le Gall [36].

### C.1   Poisson point measures

We fix $\mathcal{P}$ a probability law on some space $\Xi$.

**Definition 2.** *A (homogenous) Poisson point measure on $\mathbb{R}_{\geqslant 0} \times \Xi$, with intensity $\nu(\mathrm{d}t, \mathrm{d}\xi) = \mathrm{d}t \otimes \mathrm{d}\mathcal{P}(\xi)$, is a random measure $N$ on $\mathbb{R}_{\geqslant 0} \times \Xi$ such that*

- *For any disjoint measurable subsets $A$ and $B$ of $\mathbb{R}_{\geqslant 0} \times \Xi$, $N(A)$ and $N(B)$ are independent.*

- *For any measurable subset $A$ of $\mathbb{R}_{\geqslant 0} \times \Xi$, $N(A)$ is a Poisson random variable with parameter $\nu(A)$. (If $\nu(A) = \infty$, $N(A)$ is equal to $\infty$ almost surely.)*

**Proposition 1.** *Let $N$ be a Poisson point measure on $\mathbb{R}_{\geqslant 0} \times \Xi$ with intensity $\mathrm{d}t \otimes \mathrm{d}\mathcal{P}(\xi)$.*

*There exists a decomposition $\mathrm{d}N(t,\xi) = \sum_{k \geqslant 1} \delta_{(T_k, \xi_k)}(\mathrm{d}t, \mathrm{d}\xi)$ on $\mathbb{R}_{\geqslant 0} \times \Xi$ where $0 < T_1 < T_2 < T_3 < \ldots$ and $\xi_1, \xi_2, \xi_3, \cdots \in \Xi$ satisfy:*

- *$T_1, T_2 - T_1, T_3 - T_2, \ldots$ are i.i.d. of law exponential with rate $1$,*

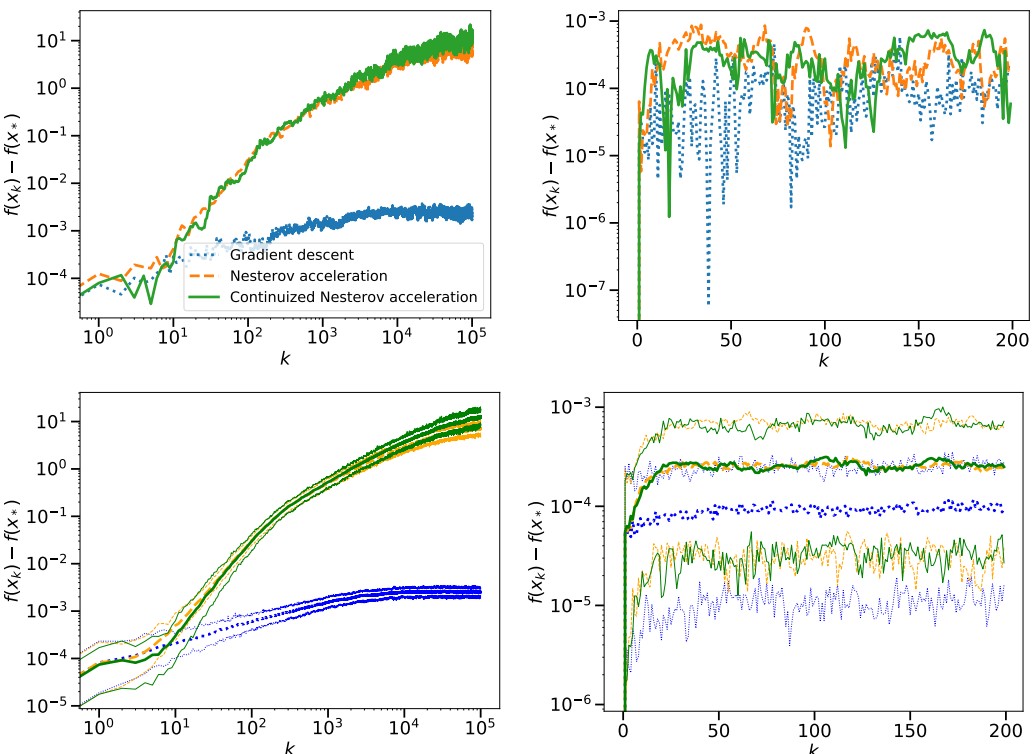

Figure 3: Effect of additive noise on gradient descent, Nesterov acceleration, and the continuized version of Nesterov acceleration, on a convex function (left) and a strongly convex function (right). All algorithms are started from the optimum $x_*$. The results shown in the above plots correspond to a single run. In the plots below, the thick line represents the average performance over $N = 100$ runs of each algorithm, while the thin lines represent the $5\%$ and $95\%$ quantiles.

- $\xi_1, \xi_2, \xi_3, \ldots$ *are i.i.d. of law $\mathcal{P}$ and independent of the $T_1, T_2, T_3, \ldots$.*

**Definition 3.** *Let $N$ be a Poisson point measure on $\mathbb{R}_{\geqslant 0} \times \Xi$ with intensity $\mathrm{d}t \otimes \mathrm{d}\mathcal{P}(\xi)$. The* filtration $\mathcal{F}_t, t \geqslant 0$, *generated by $N$ is defined by the formula*

$$\mathcal{F}_t = \sigma\left(N([0, s] \times A), \, s \leqslant t, A \subset \Xi \text{ measurable}\right).$$

## C.2 Martingales and supermartingales

Let $(\Omega, \mathcal{F}, \mathbb{P})$ be a probability space and $\mathcal{F}_t, t \geqslant 0$, a filtration on this probability space.

**Definition 4.** *A random process $x_t \in \mathbb{R}^d$, $t \geqslant 0$, is* adapted *if for all $t \geqslant 0$, $x_t$ is $\mathcal{F}_t$-measurable. An adapted process $x_t \in \mathbb{R}$, $t \geqslant 0$ is a* martingale *(resp.* supermartingale*) if for all $0 \leqslant s \leqslant t$, $\mathbb{E}[x_t \mid \mathcal{F}_s] = x_s$ (resp. $\mathbb{E}[x_t \mid \mathcal{F}_s] \leqslant x_s$).*

**Definition 5.** *A random variable $T \in [0, \infty]$ is a* stopping time *if for all $t \geqslant 0$, $\{T \leqslant t\} \in \mathcal{F}_t$.*

**Definition 6.** *A function $x_t, t \geqslant 0$, is said to be* càdlàg *if it is right continuous and for every $t > 0$, the limit $x_{t-} := \lim_{s \to t, s < t} x_s$ exists and is finite.*

**Theorem 8** (Martingale stopping theorem)**.** *Let $x_t$, $t \geqslant 0$, be a martingale (resp. supermartingale) with càdlàg trajectories and uniformly integrable. Let $T$ be a stopping time. Then $\mathbb{E}X_T = X_0$ (resp. $\mathbb{E}X_T \leqslant X_0$).*

## C.3 Stochastic ordinary differential equation with Poisson jumps

The continuized processes are the composition of an ordinary differential equation and stochastic Poisson jumps. It is thus a piecewise-deterministic Markov process [15, 16], a special case of

stochastic models that do not include any diffusion term. The stochastic calculus of this class of processes is particularly intuitive: there is no Ito correction term as with diffusive processes.

We fix $\mathcal{P}$ a probability law on some space $\Xi$, $N$ a Poisson point measure on $\mathbb{R}_{\geqslant 0} \times \Xi$ with intensity $\mathrm{d}t \otimes \mathrm{d}\mathcal{P}(\xi)$, and denote $\mathcal{F}_t$, $t \geqslant 0$, the filtration generated by $N$.

**Definition 7.** *Let $b : \mathbb{R}^d \to \mathbb{R}^d$ and $G : \mathbb{R}^d \times \Xi \to \mathbb{R}^d$ be two functions. An random process $x_t \in \mathbb{R}^d$, $t \geqslant 0$, is said to be a solution of the equation*

$$\mathrm{d}x_t = b(x_t)\mathrm{d}t + \int_\Xi G(x_t, \xi)\mathrm{d}N(t, \xi)$$

*if it is adapted, càdlàg, and for all $t \geqslant 0$,*

$$x_t = x_0 + \int_0^t b(x_s)\mathrm{d}s + \int_{[0,t]\times\Xi} G(x_{s-}, \xi)\mathrm{d}N(s, \xi) \,.$$

*If we consider the decomposition $\mathrm{d}N(t, \xi) = \sum_{k\geqslant 1} \delta_{(T_k, \xi_k)}(\mathrm{d}t, \mathrm{d}\xi)$ given by Proposition 1, then*

$$\int_{[0,t]\times\Xi} G(x_{s-}, \xi)\mathrm{d}N(s, \xi) = \sum_{k\geqslant 1} \mathbf{1}_{\{T_k \leqslant t\}} G(x_{T_k-}, \xi_k) \,.$$

Here, we consider only autonomous equations as $b$ and $G$ are a function of $x_t$, but not of $t$. However, there is no loss of generality, one can study time-dependent systems by studying the equation in the variable $(t, x_t)$. This trick is used in Appendix D.

**Proposition 2.** *Let $x_t \in \mathbb{R}^d$ be a solution of*

$$\mathrm{d}x_t = b(x_t)\mathrm{d}t + \int_\Xi G(x_t, \xi)\mathrm{d}N(t, \xi)$$

*and $\varphi : \mathbb{R}^d \to \mathbb{R}$ be a smooth function. Then*

$$\varphi(x_t) = \varphi(x_0) + \int_0^t \langle \nabla\varphi(x_s), b(x_s)\rangle \mathrm{d}s + \int_{[0,t]\times\Xi} \left(\varphi(x_{s-} + G(x_{s-}, \xi)) - \varphi(x_{s-})\right) \mathrm{d}N(s, \xi) \,.$$

*Moreover, we have the decomposition*

$$\int_{[0,t]\times\Xi} \left(\varphi(x_{s-} + G(x_{s-}, \xi)) - \varphi(x_{s-})\right) \mathrm{d}N(s, \xi)$$
$$= \int_0^t \int_\Xi \left(\varphi(x_s + G(x_s, \xi)) - \varphi(x_s)\right) \mathrm{d}t \mathrm{d}\mathcal{P}(\xi) + M_t \,,$$

*where $M_t = \int_{[0,t]\times\Xi} \left(\varphi(x_{s-} + G(x_{s-}, \xi)) - \varphi(x_{s-})\right) \left(\mathrm{d}N(s, \xi) - \mathrm{d}t \mathrm{d}\mathcal{P}(\xi)\right)$ is a martingale.*

This proposition is an elementary calculus of variations formula: to compute the value of the observable $\varphi(x_t)$, one must sum the effects of the continuous part and of the Poisson jumps. Moreover, the integral with respect to the Poisson measure $N$ becomes a martingale if the same integral with respect to its intensity measure $\mathrm{d}t \otimes \mathrm{d}\mathcal{P}(\xi)$ is removed.

## D  Analysis of the continuized Nesterov acceleration

To encompass the proofs in the convex and in the strongly convex cases in a unified way, we assume $f$ is $\mu$-strongly convex, $\mu \geqslant 0$. If $\mu > 0$, this corresponds to assuming the $\mu$-strong convexity in the usual sense; if $\mu = 0$, it means that we only assume the function to be convex. In other words, the proofs in the convex case can be obtained by taking $\mu = 0$ below.

In this section, $\mathcal{F}_t$, $t \geqslant 0$, is the filtration associated to the Poisson point measure $N$.

## D.1 Sketch of proof for Theorem 2

A complete and rigorous proof is given in Appendix D.2. Here, we only provide the heuristic of the main lines of the proof.

The proof is similar to the one of Nesterov acceleration: we prove that for some choices of parameters $\eta_t, \eta'_t, \gamma_t, \gamma'_t, t \geqslant 0$, and for some functions $A_t, B_t, t \geqslant 0$,

$$\phi_t = A_t \left( f(x_t) - f(x_*) \right) + \frac{B_t}{2} \|z_t - x_*\|^2$$

is a supermartingale. In particular, this implies that $\mathbb{E}\phi_t$ is a Lyapunov function, i.e., a non-increasing function of $t$.

To prove that $\phi_t$ is a supermartingale, it is sufficient to prove that for all infinitesimal time intervals $[t, t + \mathrm{d}t]$, $\mathbb{E}_t \phi_{t+\mathrm{d}t} \leqslant \phi_t$, where $\mathbb{E}_t$ denotes the conditional expectation knowing all the past of the Poisson process up to time $t$. Thus we would like to compute the first order variation of $\mathbb{E}_t \phi_{t+\mathrm{d}t}$. This implies computing the first order variation of $\mathbb{E}_t f(x_{t+\mathrm{d}t})$.

From (10), we see that $f(x_t)$ evolves for two reasons between $t$ and $t + \mathrm{d}t$:

- $x_t$ follows the linear ODE (8), which results in the infinitesimal variation $f(x_t) \to f(x_t) + \eta_t \langle \nabla f(x_t), z_t - x_t \rangle \mathrm{d}t$, and
- with probability $\mathrm{d}t$, $x_t$ takes a gradient step, which results in a macroscopic variation $f(x_t) \to f(x_t - \gamma_t \nabla f(x_t))$.

Combining both variations, we obtain that

$$\mathbb{E}_t f(x_{t+\mathrm{d}t}) \approx f(x_t) + \eta_t \langle \nabla f(x_t), z_t - x_t \rangle \mathrm{d}t + \mathrm{d}t \left( f(x_t - \gamma_t \nabla f(x_t)) - f(x_t) \right),$$

where the $\mathrm{d}t$ in the second term corresponds to the probability that a gradient step happens; note that the latter event is independent of the past up to time $t$.

A similar computation can be done for $\mathbb{E}_t \|z_t - x_*\|^2$. Putting things together, we obtain

$$\mathbb{E}_t \phi_{t+\mathrm{d}t} - \phi_t \approx \mathrm{d}t \Bigg( \frac{\mathrm{d}A_t}{\mathrm{d}t} (f(x_t) - f(x_*)) + A_t \eta_t \langle \nabla f(x_t), z_t - x_t \rangle$$
$$- A_t \left( f(x_t - \gamma_t \nabla f(x_t)) - f(x_t) \right) + \frac{\mathrm{d}B_t}{\mathrm{d}t} \frac{1}{2} \|z_t - x_*\|^2$$
$$+ B_t \eta'_t \langle z_t - x_*, x_t - z_t \rangle + \frac{B_t}{2} \left( \|z_t - \gamma'_t \nabla f(x_t) - x_*\|^2 - \|z_t - x_*\|^2 \right) \Bigg).$$

Using convexity and strong convexity inequalities, and a few computations, we obtain the following upper bound:

$$\mathbb{E}_t \phi_{t+\mathrm{d}t} - \phi_t \lesssim \mathrm{d}t \Bigg( \left( \frac{\mathrm{d}A_t}{\mathrm{d}t} - A_t \eta_t \right) \langle \nabla f(x_t), x_t - x_* \rangle + \left( \frac{\mathrm{d}B_t}{\mathrm{d}t} - B_t \eta'_t \right) \frac{1}{2} \|z_t - x_*\|^2$$
$$+ (A_t \eta_t - B_t \gamma'_t) \langle \nabla f(x_t), z_t - x_* \rangle + \left( B_t \eta'_t - \frac{\mathrm{d}A_t}{\mathrm{d}t} \mu \right) \frac{1}{2} \|x_t - x_*\|^2$$
$$+ \left( B_t \gamma'^2_t - A_t \gamma_t (2 - L\gamma_t) \right) \frac{1}{2} \|\nabla f(x_t)\|^2 \Bigg).$$

We want this infinitesimal variation to be non-positive. Here, we choose the parameters so that $\gamma_t = 1/L$, and all prefactors in the above expression are zero. This gives some constraints on the choices of parameters. We show that only one degree of freedom is left: the choice of the function $A_t$, that must satisfy the ODE

$$\frac{\mathrm{d}^2}{\mathrm{d}t^2} \left( \sqrt{A_t} \right) = \frac{\mu}{4L} \sqrt{A_t},$$

but whose initialization remains free. Once the initialization of the function $A_t$ is chosen, this determines the full function $A_t$ and, through the constraints, all parameters of the algorithm. As $\phi_t$ is a supermartingale (by design), a bound on the performance of the algorithm is given by

$$\mathbb{E} f(x_t) - f(x_*) \leqslant \frac{\mathbb{E}\phi_t}{A_t} \leqslant \frac{\phi_0}{A_t}.$$

The results presented in Theorem 2 correspond to one special choice of initialization for the function $A_t$.

In this sketch of proof, our derivation of the infinitesimal variation is intuitive and elementary; however it can be made more rigorous and concise—albeit more technical—using classical results from stochastic calculus, namely Proposition 2. This is our approach in Appendix D.2.

## D.2   Noiseless case: proofs of Theorem 2 and of the bounds of Theorem 3

In this section, we analyze the convergence of the continuized iteration (10)-(11), that we recall for the reader's convenience:

$$\mathrm{d}x_t = \eta_t(z_t - x_t)\mathrm{d}t - \gamma_t \nabla f(x_t)\mathrm{d}N(t)\,,$$
$$\mathrm{d}z_t = \eta_t'(x_t - z_t)\mathrm{d}t - \gamma_t'\nabla f(x_t)\mathrm{d}N(t)\,.$$

The choices of parameters $\eta_t, \eta_t', \gamma_t, \gamma_t'$, $t \geqslant 0$, and the corresponding convergence bounds follow naturally from the analysis. We seek sufficient conditions under which the function

$$\phi_t = A_t\left(f(x_t) - f_*\right) + \frac{B_t}{2}\|z_t - x_*\|^2$$

is a supermartingale.

The process $\bar{x}_t = (t, x_t, z_t)$ satisfies the equation

$$\mathrm{d}\bar{x}_t = b(\bar{x}_t)\mathrm{d}t + G(\bar{x}_t)\mathrm{d}N(t)\,, \quad b(\bar{x}_t) = \begin{pmatrix} 1 \\ \eta_t(z_t - x_t) \\ \eta_t'(x_t - z_t) \end{pmatrix}\,, \quad G(\bar{x}_t) = \begin{pmatrix} 0 \\ -\gamma_t \nabla f(x_t) \\ -\gamma_t' \nabla f(x_t) \end{pmatrix}\,.$$

We thus apply Proposition 2 to $\phi_t = \varphi(\bar{x}_t) = \varphi(t, x_t, z_t)$ where

$$\varphi(t, x, z) = A_t\left(f(x) - f(x_*)\right) + \frac{B_t}{2}\|z - x_*\|^2\,,$$

we obtain:

$$\phi_t = \phi_0 + \int_0^t \langle \nabla\varphi(\bar{x}_s), b(\bar{x}_s)\rangle \mathrm{d}s + \int_0^t \left(\varphi(\bar{x}_s + G(\bar{x}_s)) - \varphi(\bar{x}_s)\right)\mathrm{d}s + M_t\,,$$

where $M_t$ is a martingale. Thus, to show that $\varphi_t$ is a supermartingale, it is sufficient to show that the map $t \mapsto \int_0^t \langle \nabla\varphi(\bar{x}_s), b(\bar{x}_s)\rangle \mathrm{d}s + \int_0^t \left(\varphi(\bar{x}_s + G(\bar{x}_s)) - \varphi(\bar{x}_s)\right)\mathrm{d}s$ is non-increasing almost surely, i.e.,

$$I_t := \langle \nabla\varphi(\bar{x}_t), b(\bar{x}_t)\rangle + \varphi(\bar{x}_t + G(\bar{x}_t)) - \varphi(\bar{x}_t) \leqslant 0\,.$$

We now compute

$$\langle \nabla\varphi(\bar{x}_t), b(\bar{x}_t)\rangle = \partial_t\varphi(\bar{x}_t) + \langle \partial_x\varphi(\bar{x}_t), \eta_t(z_t - x_t)\rangle + \langle \partial_z\varphi(\bar{x}_t), \eta_t'(x_t - z_t)\rangle$$
$$= \frac{\mathrm{d}A_t}{\mathrm{d}t}\left(f(x_t) - f(x_*)\right) + \frac{\mathrm{d}B_t}{\mathrm{d}t}\frac{1}{2}\|z_t - x_*\|^2 + A_t\eta_t\langle \nabla f(x_t), z_t - x_t\rangle$$
$$+ B_t\eta_t'\langle z_t - x_*, x_t - z_t\rangle\,.$$

Here, we use that as $f$ is $\mu$-strongly convex,

$$f(x_t) - f(x_*) \leqslant \langle \nabla f(x_t), x_t - x_*\rangle - \frac{\mu}{2}\|x_t - x_*\|^2\,,$$

and the simple bound

$$\langle z_t - x_*, x_t - z_t\rangle = \langle z_t - x_*, x_t - x_*\rangle - \|z_t - x_*\|^2 \leqslant \|z_t - x_*\|\|x_t - x_*\| - \|z_t - x_*\|^2$$
$$\leqslant \frac{1}{2}\left(\|z_t - x_*\|^2 + \|x_t - x_*\|^2\right) - \|z_t - x_*\|^2 = \frac{1}{2}\left(\|x_t - x_*\|^2 - \|z_t - x_*\|^2\right)\,.$$

This gives

$$\langle \nabla\varphi(\bar{x}_t), b(\bar{x}_t)\rangle \leqslant \left(\frac{\mathrm{d}A_t}{\mathrm{d}t} - A_t\eta_t\right)\langle \nabla f(x_t), x_t - x_*\rangle + \left(B_t\eta_t' - \frac{\mathrm{d}A_t}{\mathrm{d}t}\mu\right)\frac{1}{2}\|x_t - x_*\|^2 \quad (29)$$

$$+ \left(\frac{\mathrm{d}B_t}{\mathrm{d}t} - B_t\eta_t'\right)\frac{1}{2}\|z_t - x_*\|^2 + A_t\eta_t\langle \nabla f(x_t), z_t - x_*\rangle\,. \quad (30)$$

Further,
$$\varphi(\bar{x}_t + G(\bar{x}_t)) - \varphi(\bar{x}_t) = A_t \left( f(x_t - \gamma_t \nabla f(x_t)) - f(x_t) \right)$$
$$+ \frac{B_t}{2} \left( \|(z_t - x_*) - \gamma_t' \nabla f(x_t)\|^2 - \|z_t - x_*\|^2 \right) .$$

As $f$ is $L$-smooth,
$$f(x_t - \gamma_t \nabla f(x_t)) - f(x_t) \leqslant \langle \nabla f(x_t), -\gamma_t \nabla f(x_t) \rangle + \frac{L}{2} \|\gamma_t \nabla f(x_t)\|^2$$
$$= -\gamma_t \left( 2 - L\gamma_t \right) \frac{1}{2} \|\nabla f(x_t)\|^2 .$$

This gives
$$\varphi(\bar{x}_t + G(\bar{x}_t)) - \varphi(\bar{x}_t) \leqslant \left( B_t \gamma_t'^2 - A_t \gamma_t \left( 2 - L\gamma_t \right) \right) \frac{1}{2} \|\nabla f(x_t)\|^2 - B_t \gamma_t' \langle \nabla f(x_t), z_t - x_* \rangle . \tag{31}$$

Finally, combining (29)-(30) with (31), we obtain
$$I_t \leqslant \left( \frac{\mathrm{d}A_t}{\mathrm{d}t} - A_t \eta_t \right) \langle \nabla f(x_t), x_t - x_* \rangle + \left( \frac{\mathrm{d}B_t}{\mathrm{d}t} - B_t \eta_t' \right) \frac{1}{2} \|z_t - x_*\|^2 \tag{32}$$
$$+ \left( A_t \eta_t - B_t \gamma_t' \right) \langle \nabla f(x_t), z_t - x_* \rangle + \left( B_t \eta_t' - \frac{\mathrm{d}A_t}{\mathrm{d}t} \mu \right) \frac{1}{2} \|x_t - x_*\|^2 \tag{33}$$
$$+ \left( B_t \gamma_t'^2 - A_t \gamma_t \left( 2 - L\gamma_t \right) \right) \frac{1}{2} \|\nabla f(x_t)\|^2 . \tag{34}$$

Remember that $I_t \leqslant 0$ is a sufficient condition for $\phi_t$ to be a supermartingale. Here, we choose the parameters $\eta_t, \eta_t', \gamma_t, \gamma_t', t \geqslant 0$, so that all prefactors are 0. We start by taking $\gamma_t \equiv \frac{1}{L}$ (other choices $\gamma_t < \frac{2}{L}$ could be possible but would give similar results) and we want to satisfy
$$\frac{\mathrm{d}A_t}{\mathrm{d}t} = A_t \eta_t , \qquad \frac{\mathrm{d}B_t}{\mathrm{d}t} = B_t \eta_t' \qquad A_t \eta_t = B_t \gamma_t' , \qquad B_t \eta_t' = \frac{\mathrm{d}A_t}{\mathrm{d}t} \mu , \qquad B_t \gamma_t'^2 = \frac{A_t}{L} .$$

To satisfy the last equation, we choose
$$\gamma_t' = \sqrt{\frac{A_t}{LB_t}} . \tag{35}$$

To satisfy the third equation, we choose
$$\eta_t = \frac{B_t \gamma_t'}{A_t} = \sqrt{\frac{2B_t}{LA_t}} . \tag{36}$$

To satisfy the fourth equation, we choose
$$\eta_t' = \frac{\mathrm{d}A_t}{\mathrm{d}t} \frac{\mu}{B_t} = \frac{A_t \eta_t \mu}{B_t} = \mu \sqrt{\frac{A_t}{LB_t}} . \tag{37}$$

Having now all parameters $\eta_t, \eta_t', \gamma_t, \gamma_t'$ constrained, we now have that $\phi_t$ is Lyapunov if
$$\frac{\mathrm{d}A_t}{\mathrm{d}t} = A_t \eta_t = \sqrt{\frac{A_t B_t}{L}} , \qquad\qquad \frac{\mathrm{d}B_t}{\mathrm{d}t} = B_t \eta_t' = \mu \sqrt{\frac{A_t B_t}{L}} .$$

This only leaves the choice of the initialization $(A_0, B_0)$ as free: both the algorithm and the Lyapunov depend on it. (Actually, only the relative value $A_0/B_0$ matters.) Instead of solving the above system of two coupled non-linear ODEs, it is convenient to turn them into a single second-order linear ODE:
$$\frac{\mathrm{d}}{\mathrm{d}t} \left( \sqrt{A_t} \right) = \frac{1}{2\sqrt{A_t}} \frac{\mathrm{d}A_t}{\mathrm{d}t} = \frac{1}{2} \sqrt{\frac{B_t}{L}} , \qquad \frac{\mathrm{d}}{\mathrm{d}t} \left( \sqrt{B_t} \right) = \frac{1}{2\sqrt{B_t}} \frac{\mathrm{d}B_t}{\mathrm{d}t} = \frac{\mu}{2} \sqrt{\frac{A_t}{L}} . \tag{38}$$

This can also be restated as
$$\frac{\mathrm{d}^2}{\mathrm{d}t^2} \left( \sqrt{A_t} \right) = \frac{\mu}{4L} \sqrt{A_t} , \qquad\qquad \sqrt{B_t} = 2\sqrt{L} \frac{\mathrm{d}}{\mathrm{d}t} \left( \sqrt{A_t} \right) . \tag{39}$$

### D.2.1 Proof of the first part (convex case)

We now assume $\mu = 0$, and we choose the solution such that $A_0 = 0$ and $B_0 = 1$. From (38), we have $\frac{\mathrm{d}}{\mathrm{d}t}\left(\sqrt{B_t}\right) = 0$, thus $B_t \equiv 1$, and $\frac{\mathrm{d}}{\mathrm{d}t}\left(\sqrt{A_t}\right) = \frac{1}{2\sqrt{L}}$, thus $\sqrt{A_t} = \frac{t}{2\sqrt{L}}$. The parameters of the algorithm are given by (35)-(37): $\eta_t = \frac{2}{t}$, $\eta_t' = 0$, $\gamma_t' = \frac{t}{2L}$ (and we had chosen $\gamma_t = \frac{1}{L}$).

From the fact that $\phi_t$ is a supermartingale, we obtain that the associated algorithm satisfies

$$\mathbb{E}f(x_t) - f(x_*) \leqslant \frac{\mathbb{E}\phi_t}{A_t} \leqslant \frac{\phi_0}{A_t} = \frac{2L\|z_0 - x_*\|^2}{t^2}\,.$$

This proves the first part of Theorem 2.

Further, one can apply martingale stopping Theorem 8 to the supermartingale $\phi_t$ with the stopping time $T_k$ to obtain

$$\mathbb{E}\left[A_{T_k}\left(f(\tilde{x}_k) - f(x_*)\right)\right] = \mathbb{E}\left[A_{T_k}\left(f(x_{T_k}) - f(x_*)\right)\right] \leqslant \mathbb{E}\phi_{T_k} \leqslant \phi_0 = \|z_0 - x_*\|^2\,.$$

This proves the formula of Theorem 3.1.

### D.2.2 Proof of the second part (strongly convex case)

We now assume $\mu > 0$. We consider the solution of (39) that is exponential:

$$\sqrt{A_t} = \sqrt{A_0}\exp\left(\frac{1}{2}\sqrt{\frac{\mu}{L}}t\right)\,, \qquad \sqrt{B_t} = \sqrt{A_0}\sqrt{\mu}\exp\left(\frac{1}{2}\sqrt{\frac{\mu}{L}}t\right)\,.$$

The parameters of the algorithm are given by (35)-(37): $\eta_t = \eta_t' = \sqrt{\frac{\mu}{L}}$, $\gamma_t' = \frac{1}{\sqrt{\mu L}}$ (and we had chosen $\gamma_t = \frac{1}{L}$).

From the fact that $\phi_t$ is a supermartingale, we obtain that the associated algorithm satisfies

$$\mathbb{E}f(x_t) - f(x_*) \leqslant \frac{\mathbb{E}\phi_t}{A_t} \leqslant \frac{\phi_0}{A_t} = \frac{A_0(f(x_0) - f(x_*)) + A_0\frac{\mu}{2}\|z_0 - x_*\|^2}{A_t}$$

$$= \left(f(x_0) - f(x_*) + \frac{\mu}{2}\|z_0 - x_*\|^2\right)\exp\left(-\sqrt{\frac{\mu}{L}}t\right)\,.$$

This proves the second part of Theorem 2. Similarly to above, one can also apply the martingale stopping theorem to prove the formula of Theorem 3.2.

**Remark 2.** *In the above derivation, in both the convex and strongly convex cases, we choose a particular solution of* (39), *while several solutions are possible. In the convex case, we make the choice $A_0 = 0$ to have a succinct bound that does not depend on $f(x_0) - f(x_*)$. More importantly, in the strongly convex case, we choose the solution that satisfies the relation $\sqrt{\mu}\sqrt{A_t} = \sqrt{B_t}$, which implies that $\eta_t, \eta_t', \gamma_t'$, are constant functions of $t$, and $\eta_t = \eta_t'$. These conditions help solving in closed form the continuous part of the process*

$$\mathrm{d}x_t = \eta_t(z_t - x_t)\mathrm{d}t\,,$$
$$\mathrm{d}z_t = \eta_t'(x_t - z_t)\mathrm{d}t\,,$$

*which is crucial if we want to have a discrete implementation of our method (for more details, see Theorem 3 and its proof). However, in the strongly convex case, considering other solutions would be interesting, for instance to have an algorithm converging to the convex one as $\mu \to 0$.*

### D.3 With additive noise: proof of Theorem 7

The proof of this theorem is along the same lines as the proof of Theorem 2 above. Here, we only give the major differences.

We analyze the convergence of the continuized stochastic iteration (15)-(16), that we recall for the reader's convenience:

$$\mathrm{d}x_t = \eta_t(z_t - x_t)\mathrm{d}t - \gamma_t\int_\Xi \nabla f(x_t, \xi)\mathrm{d}N(t, \xi)\,,$$

$$\mathrm{d}z_t = \eta_t'(x_t - z_t)\mathrm{d}t - \gamma_t'\int_\Xi \nabla f(x_t, \xi)\mathrm{d}N(t, \xi)\,.$$

In this setting, we loose the property that

$$\phi_t = A_t \left( f(x_t) - f_* \right) + \frac{B_t}{2} \| z_t - x_* \|^2$$

is a supermartingale. However, we bound the increase of $\phi_t$.

The process $\bar{x}_t = (t, x_t, z_t)$ satisfies the equation

$$\mathrm{d}\bar{x}_t = b(\bar{x}_t)\mathrm{d}t + \int_\Xi G(\bar{x}_t, \xi)\mathrm{d}N(t, \xi), \quad b(\bar{x}_t) = \begin{pmatrix} 1 \\ \eta_t(z_t - x_t) \\ \eta_t'(x_t - z_t) \end{pmatrix}, \quad G(\bar{x}_t, \xi) = \begin{pmatrix} 0 \\ -\gamma_t \nabla f(x_t, \xi) \\ -\gamma_t' \nabla f(x_t, \xi) \end{pmatrix}.$$

We apply Proposition 2 to $\phi_t = \varphi(\bar{x}_t) = \varphi(t, x_t, z_t)$ and obtain

$$\phi_t = \phi_0 + \int_0^t I_s \mathrm{d}s + M_t, \tag{40}$$

where $M_t$ is a martingale and

$$I_t = \langle \nabla \varphi(\bar{x}_t), b(\bar{x}_t) \rangle + \mathbb{E}_\xi \varphi(\bar{x}_t + G(\bar{x}_t, \xi)) - \varphi(\bar{x}_t).$$

The computation of the first term remains the same: the inequality (29)-(30) holds. The computation of the second term becomes

$$\mathbb{E}_\xi \varphi(\bar{x}_t + G(\bar{x}_t, \xi)) - \varphi(\bar{x}_t) = A_t \left( \mathbb{E}_\xi f(x_t - \gamma_t \nabla f(x_t, \xi)) - f(x_t) \right)$$
$$+ \frac{B_t}{2} \left( \mathbb{E}_\xi \| (z_t - x_*) - \gamma_t' \nabla f(x_t, \xi) \|^2 - \| z_t - x_* \|^2 \right).$$

As $f$ is $L$-smooth,

$$f(x_t - \gamma_t \nabla f(x_t, \xi)) - f(x_t) \leqslant \langle \nabla f(x_t), -\gamma_t \nabla f(x_t, \xi) \rangle + \frac{L}{2} \| \gamma_t \nabla f(x_t, \xi) \|^2,$$

$$\mathbb{E}_\xi f(x_t - \gamma_t \nabla f(x_t, \xi)) - f(x_t) \leqslant \langle \nabla f(x_t), -\gamma_t \mathbb{E}_\xi \nabla f(x_t, \xi) \rangle + \frac{L}{2} \mathbb{E}_\xi \| \gamma_t \nabla f(x_t, \xi) \|^2.$$

By assumptions (27) and (28), the stochastic gradient $\nabla f(x, \xi)$ is unbiased and has a variance bounded by $\sigma^2$, which implies $\mathbb{E}_\xi \| \nabla f(x_t, \xi) \|^2 \leqslant \| \nabla f(x_t) \|^2 + \sigma^2$. Thus

$$\mathbb{E}_\xi f(x_t - \gamma_t \nabla f(x_t, \xi)) - f(x_t) \leqslant -\gamma_t (2 - L\gamma_t) \frac{1}{2} \| \nabla f(x_t) \|^2 + \sigma^2 \frac{L\gamma_t^2}{2}.$$

Similarly,

$$\mathbb{E}_\xi \| (z_t - x_*) - \gamma_t' \nabla f(x_t, \xi) \|^2 - \| z_t - x_* \|^2 = -2\gamma_t' \langle \mathbb{E}_\xi \nabla f(x_t, \xi), z_t - x_* \rangle + \gamma_t'^2 \mathbb{E}_\xi \| \nabla f(x_t, \xi) \|^2$$
$$\leqslant -2\gamma_t' \langle \nabla f(x_t), z_t - x_* \rangle + \gamma_t'^2 \| \nabla f(x_t) \|^2 + \sigma^2 \gamma_t'^2.$$

This gives

$$\varphi(\bar{x}_t + G(\bar{x}_t)) - \varphi(\bar{x}_t) \leqslant \left( B_t \gamma_t'^2 - A_t \gamma_t (2 - L\gamma_t) \right) \frac{1}{2} \| \nabla f(x_t) \|^2 - B_t \gamma_t' \langle \nabla f(x_t), z_t - x_* \rangle$$
$$+ \frac{\sigma^2}{2} \left( A_t L \gamma_t^2 + B_t \gamma_t'^2 \right).$$

Combining the bounds, we obtain

$$I_t \leqslant \left( \frac{\mathrm{d}A_t}{\mathrm{d}t} - A_t \eta_t \right) \langle \nabla f(x_t), x_t - x_* \rangle + \left( \frac{\mathrm{d}B_t}{\mathrm{d}t} - B_t \eta_t' \right) \frac{1}{2} \| z_t - x_* \|^2$$
$$+ (A_t \eta_t - B_t \gamma_t') \langle \nabla f(x_t), z_t - x_* \rangle + \left( B_t \eta_t' - \frac{\mathrm{d}A_t}{\mathrm{d}t} \mu \right) \frac{1}{2} \| x_t - x_* \|^2$$
$$+ \left( B_t \gamma_t'^2 - A_t \gamma_t (2 - L\gamma_t) \right) \frac{1}{2} \| \nabla f(x_t) \|^2 + \frac{\sigma^2}{2} \left( A_t L \gamma_t^2 + B_t \gamma_t'^2 \right),$$

which is an additive perturbation of the bound (32)-(34) in the noiseless case, with a perturbation proportional to $\sigma^2$. The choices of parameters of Theorem 2 cancel all first five prefactors, and satisfy $\gamma_t = \frac{1}{L}$, $A_t L \gamma_t^2 = B_t \gamma_t'^2$. We thus obtain

$$I_t \leqslant \sigma^2 \frac{A_t}{L}.$$

This bound controls the increase of $\phi_t$. Using the decomposition (50), we obtain

$$\mathbb{E}f(x_t) - f(x_*) \leqslant \frac{\mathbb{E}\phi_t}{A_t} \leqslant \frac{\phi_0}{A_t} + \frac{\int_0^t \mathbb{E}I_s \mathrm{d}s}{A_t}$$

$$\leqslant \frac{A_0(f(x_0) - f(x_*)) + B_0\|z_0 - x_*\|^2}{A_t} + \frac{\sigma^2}{L}\frac{\int_0^t A_s \mathrm{d}s}{A_t} \, .$$

### D.3.1    Proof of the first part (convex case)

In this case, $A_t = \frac{t^2}{2L}$ and $B_0 = 1$. Thus $\int_0^t A_s \mathrm{d}s = \frac{1}{2L}\frac{t^3}{3}$. Thus

$$\mathbb{E}f(x_t) - f(x_*) \leqslant \frac{2L\|z_0 - x_*\|^2}{t^2} + \sigma^2\frac{t}{3L} \, .$$

### D.3.2    Proof of the second part (strongly convex case)

In this case, $A_t = A_0 \exp\left(\sqrt{\frac{\mu}{L}}t\right)$ and $B_0 = A_0\frac{\mu}{2}$. Thus $\int_0^t A_s \mathrm{d}s \leqslant A_0\sqrt{\frac{\mu}{L}}^{-1} \exp\left(\sqrt{\frac{\mu}{L}}t\right) = \sqrt{\frac{L}{\mu}}A_t$. Thus

$$\mathbb{E}f(x_t) - f(x_*) \leqslant \left(f(x_0) - f(x_*) + \frac{\mu}{2}\|z_0 - x_*\|^2\right)\exp\left(-\sqrt{\frac{\mu}{L}}t\right) + \sigma^2\frac{1}{\sqrt{\mu L}} \, .$$

### D.4    With Pure Multiplicative Noise: Proof of Theorem 4

The proof of this theorem mimics the proof of Theorem 2, with a slightly different Lyapunov function. We recall that in Section 5, the function $f$ is of the form:

$$\forall x \in \mathbb{R}^d, f(x) = \mathbb{E}\left[\frac{1}{2}(\langle a, x\rangle - b)^2\right],$$

where $\xi = (a, b) \in \mathbb{R}^d \times \mathbb{R}$ is of law $\mathcal{P}$. Thanks to the *noiseless assumption*, for $H = \mathbb{E}\left[aa^\top\right]$, we also have:

$$\forall x \in \mathbb{R}^d, f(x) = \frac{1}{2}\|x - x_*\|_H^2.$$

The Lyapunov function studied in the proof of Theorem 2 would then write as, for $t \in \mathbb{R}_{\geqslant 0}$:

$$\phi_t = \frac{A_t}{2}\|x_t - x_*\|_H^2 + \frac{B_t}{2}\|z_t - x_*\|^2.$$

An acceleration of stochastic gradient descent using this Lyapunov function has been done by Vaswani et al. [52]. In order to have an analysis similar to Nesterov acceleration, the authors make a strong growth condition, which is too strong for many stochastic gradient problems and for our application to gossip algorithms. Instead, our analysis requires a bounded statistical condition number $\tilde{\kappa}$, and performs a shift in terms of dependency over $H$: $\|x - x_*\|_H^2$ becomes $\|x - x_*\|^2$, and $\|z_t - x_*\|^2$ becomes $\|z_t - x_*\|_{H^{-1}}^2$. The new Lyapunov function writes:

$$\phi_t = \frac{A_t}{2}\|x_t - x_*\|^2 + \frac{B_t}{2}\|z_t - x_*\|_{H^{-1}}^2.$$

As in Theorem 2, the proof consists in proving that for carefully chosen parameters, $\phi_t$ is a super-matingale. The process $\bar{x}_t = (t, x_t, z_t)$ satisfies the equation

$$\mathrm{d}\bar{x}_t = b(\bar{x}_t)\mathrm{d}t + \int_\Xi G(\bar{x}_t, \xi)\mathrm{d}N(t, \xi), \quad b(\bar{x}_t) = \begin{pmatrix}1\\ \eta_t(z_t - x_t)\\ \eta_t'(x_t - z_t)\end{pmatrix}, \quad G(\bar{x}_t, \xi) = \begin{pmatrix}0\\ -\gamma_t\nabla f(x_t, \xi)\\ -\gamma_t'\nabla f(x_t, \xi)\end{pmatrix}.$$

We apply Proposition 2 to $\phi_t = \varphi(\bar{x}_t) = \varphi(t, x_t, z_t)$ and obtain:

$$\phi_t = \phi_0 + \int_0^t I_s \mathrm{d}s + M_t \, ,$$

where $M_t$ is a martingale and
$$I_t = \langle \nabla\varphi(\bar{x}_t), b(\bar{x}_t)\rangle + \mathbb{E}_\xi\varphi(\bar{x}_t + G(\bar{x}_t, \xi)) - \varphi(\bar{x}_t).$$

Since the Lyapunov function is not the same, we need to explicit here each term. The first term writes:

$$\langle \nabla\varphi(\bar{x}_t), b(\bar{x}_t)\rangle = \frac{1}{2}\frac{\mathrm{d}A_t}{\mathrm{d}t}\|x_t - x_*\|^2 + \frac{1}{2}\frac{\mathrm{d}B_t}{\mathrm{d}t}\|z_t - x_*\|_{H^{-1}}^2$$
$$+ A_t\eta_t\langle x_t - x_*, z_t - x_t\rangle + B_t\eta'_t\langle H^{-1}(z_t - x_*), x_t - z_t\rangle.$$

Mimicking the proof of Theorem 2, we write
$$\frac{1}{2}\|x_t - x_*\|^2 \leqslant \|x_t - x_*\|^2 - \frac{\mu}{2}\|x_t - x_*\|_{H^{-1}}^2,$$

and

$$\langle H^{-1}(z_t - x_*), x_t - z_t\rangle = \langle z_t - x_*, x_t - x_*\rangle_{H^{-1}} - \|z_t - x_*\|_{H^{-1}}^2$$
$$\leqslant \frac{1}{2}\left(\|x_t - x_*\|_{H^{-1}}^2 - \|z_t - x_*\|_{H^{-1}}^2\right).$$

Hence:

$$\langle \nabla\varphi(\bar{x}_t), b(\bar{x}_t)\rangle \leqslant \frac{\mathrm{d}A_t}{\mathrm{d}t}\|x_t - x_*\|^2 + \left(B_t\eta'_t - \frac{\mathrm{d}A_t}{\mathrm{d}t}\mu\right)\frac{1}{2}\|x_t - x_*\|_{H^{-1}}^2$$
$$+ \left(\frac{\mathrm{d}B_t}{\mathrm{d}t} - B_t\eta'_t\right)\frac{1}{2}\|z_t - x_*\|_{H^{-1}}^2 + A_t\eta_t\langle x_t - x_*, z_t - x_t\rangle.$$

Further,

$$\varphi(\bar{x}_t + G(\bar{x}_t)) - \varphi(\bar{x}_t) = \frac{A_t}{2}\left(\|x_t - \gamma_t\nabla f(x_t, \xi) - x_*\|^2 - \|x_t - x_*\|^2\right)$$
$$+ \frac{B_t}{2}\left(\|(z_t - x_*) - \gamma'_t\nabla f(x_t, \xi)\|_{H^{-1}}^2 - \|z_t - x_*\|_{H^{-1}}^2\right).$$

Then, expanding and taking expectation over $\xi$ of the first term:

$$\mathbb{E}_\xi\left[\frac{1}{2}\|x_t - \gamma_t\nabla f(x_t, \xi) - x_*\|^2 - \frac{1}{2}\|x_t - x_*\|^2\right] = \frac{\gamma_t^2}{2}\mathbb{E}_\xi\left[\|\nabla f(x_t, \xi)\|^2\right] - \gamma_t\langle H(x_t - x_*), x_t - x_*\rangle$$
$$\leqslant \left(\frac{R^2\gamma_t^2}{2} - \gamma_t\right)\|x_t - x_*\|_H^2,$$

where we used the definition of $R^2$ in Equation (19):

$$\mathbb{E}_\xi\left[\|\nabla f(x_t, \xi)\|^2\right] = (x_t - x_*)^\top\mathbb{E}\left[aa^\top aa^\top\right](x_t - x_*)$$
$$= (x_t - x_*)^\top\mathbb{E}\left[\|a\|^2 aa^\top\right](x_t - x_*)$$
$$\leqslant R^2(x_t - x_*)^\top H(x_t - x_*).$$

The second term writes:

$$\frac{1}{2}\mathbb{E}_\xi\left[\|(z_t - x_*) - \gamma'_t\nabla f(x_t, \xi)\|_{H^{-1}}^2 - \|z_t - x_*\|_{H^{-1}}^2\right] = \frac{\gamma'^2_t}{2}\mathbb{E}_\xi\left[\|\nabla f(x_t, \xi)\|_{H^{-1}}^2\right]$$
$$- \gamma'_t\langle x_t - x_*, z_t - x_*\rangle$$
$$\leqslant \frac{\tilde{\kappa}\gamma'^2_t}{2}\|x_t - x_*\|_H^2$$
$$- \gamma'_t\langle x_t - x_*, z_t - x_*\rangle,$$

where we used the definition of $\tilde{\kappa}$ in Equation (20):

$$\mathbb{E}_\xi\left[\|\nabla f(x_t, \xi)\|_{H^{-1}}^2\right] = (x_t - x_*)^\top\mathbb{E}\left[aa^\top H^{-1}aa^\top\right](x_t - x_*)$$
$$= (x_t - x_*)^\top\mathbb{E}\left[a\|a\|_{H^{-1}}^2 a^\top\right](x_t - x_*)$$
$$\leqslant \tilde{\kappa}(x_t - x_*)^\top H(x_t - x_*).$$

Combining these inequalities gives the following upper-bound on $I_t$:

$$I_t \leqslant \left( \frac{\mathrm{d}A_t}{\mathrm{d}t} - A_t \eta_t \right) \|x_t - x_*\|^2 + \left( \frac{\mathrm{d}B_t}{\mathrm{d}t} - B_t \eta_t' \right) \frac{1}{2} \|z_t - x_*\|_{H^{-1}}^2$$

$$+ (A_t \eta_t - B_t \gamma_t') \langle x_t - x_*, z_t - x_* \rangle + \left( B_t \eta_t' - \frac{\mathrm{d}A_t}{\mathrm{d}t} \mu \right) \frac{1}{2} \|x_t - x_*\|_{H^{-1}}^2$$

$$+ \left( \tilde{\kappa} B_t \gamma_t'^2 - A_t \gamma_t \left( 2 - R^2 \gamma_t \right) \right) \frac{1}{2} \|x_t - x_*\|_H^2$$

Since $I_t \leqslant 0$ is still a sufficient condition for $\phi_t$ to be a supermartingale, we choose parameters such that all prefactors are equal to 0. We first take $\gamma_t = \frac{1}{R^2}$, and we want to satisfy:

$$\frac{\mathrm{d}A_t}{\mathrm{d}t} = A_t \eta_t, \qquad \frac{\mathrm{d}B_t}{\mathrm{d}t} = B_t \eta_t' \qquad A_t \eta_t = B_t \gamma_t', \qquad B_t \eta_t' = \frac{\mathrm{d}A_t}{\mathrm{d}t} \mu, \qquad B_t \gamma_t'^2 = \frac{A_t}{\tilde{\kappa} R^2}.$$

To satisfy that last equality, we choose:

$$\gamma_t' = \sqrt{\frac{A_t}{B_t \tilde{\kappa} R^2}}.$$

The rest of the proof then follows just as in the proof of Theorem D.2.

# E  Proof of Theorem 3

By integrating the ODE

$$\mathrm{d}x_t = \eta_t (z_t - x_t) \mathrm{d}t,$$
$$\mathrm{d}z_t = \eta_t' (x_t - z_t) \mathrm{d}t,$$

between $T_k$ and $T_{k+1}-$, we obtain that there exists $\tau_k, \tau_k''$, such that

$$\tilde{y}_k = x_{T_{k+1}-} = x_{T_k} + \tau_k (z_{T_k} - x_{T_k}) = \tilde{x}_k + \tau_k (\tilde{z}_k - \tilde{x}_k), \qquad (41)$$

$$z_{T_{k+1}-} = z_{T_k} + \tau_k'' (x_{T_k} - z_{T_k}) = \tilde{z}_k + \tau_k'' (\tilde{x}_k - \tilde{z}_k).$$

From the first equation, we have $\tilde{x}_k = \frac{1}{1-\tau_k} (\tilde{y}_k - \tau_k \tilde{z}_k)$, which gives by substitution in the second equation,

$$z_{T_{k+1}-} = \tilde{z}_k + \tau_k'' \left( \frac{1}{1-\tau_k} (\tilde{y}_k - \tau_k \tilde{z}_k) - \tilde{z}_k \right)$$

$$= \tilde{z}_k + \tau_k' (\tilde{y}_k - \tilde{z}_k),$$

where $\tau_k' = \frac{\tau_k''}{1-\tau_k}$.

Further, from (6)-(7), we obtain the equations

$$\tilde{x}_{k+1} = x_{T_{k+1}} = x_{T_{k+1}-} - \gamma_{T_{k+1}} \nabla f(x_{T_{k+1}-}) = \tilde{y}_k - \gamma_{T_{k+1}} \nabla f(\tilde{y}_k), \qquad (42)$$

$$\tilde{z}_{k+1} = z_{T_{k+1}} = z_{T_{k+1}-} - \gamma_{T_{k+1}}' \nabla f(x_{T_{k+1}-}) = \tilde{z}_k + \tau_k' (\tilde{y}_k - \tilde{z}_k) - \gamma_{T_{k+1}}' \nabla f(\tilde{y}_k). \qquad (43)$$

The stated equation (12)-(14) are the combination of (41), (42) and (43).

1. The parameters of Theorem 2.(1) are $\eta_t = \frac{2}{t}, \eta_t' = 0, \gamma_t = \frac{1}{L}$ and $\gamma_t' = \frac{t}{2L}$. In this case, the ODE

$$\mathrm{d}x_t = \eta_t (z_t - x_t) \mathrm{d}t = \frac{2}{t} (z_t - x_t) \mathrm{d}t,$$

$$\mathrm{d}z_t = \eta_t' (x_t - z_t) \mathrm{d}t = 0,$$

can be integrated in closed form: for $t \geqslant t_0$,

$$x_t = z_{t_0} + \left( \frac{t_0}{t} \right)^2 (x_{t_0} - z_{t_0}) = x_{t_0} + \left( 1 - \left( \frac{t_0}{t} \right)^2 \right) (z_{t_0} - x_{t_0}),$$

$$z_t = z_{t_0}.$$

In particular, taking $t_0 = T_k$, $t = T_{k+1}-$, we obtain $\tau_k = 1 - \left( \frac{T_k}{T_{k+1}} \right)^2$, $\tau_k'' = 0$ and thus $\tau_k' = \frac{\tau_k''}{1-\tau_k} = 0$. Finally, $\tilde{\gamma}_k = \gamma_{T_k} = \frac{1}{L}$ and $\tilde{\gamma}_k' = \gamma_{T_k}' = \frac{T_k}{2L}$.

2. The parameters of Theorem [2].(2) are $\eta_t = \eta_t' \equiv \sqrt{\frac{\mu}{L}}, \gamma_t \equiv \frac{1}{L}$ and $\gamma_t' \equiv \frac{1}{\sqrt{\mu L}}$. In this case, the ODE

$$\mathrm{d}x_t = \eta_t(z_t - x_t)\mathrm{d}t = \sqrt{\frac{\mu}{L}}(z_t - x_t)\mathrm{d}t\,,$$

$$\mathrm{d}z_t = \eta_t'(x_t - z_t)\mathrm{d}t = \sqrt{\frac{\mu}{L}}(x_t - z_t)\mathrm{d}t\,,$$

can also be integrated in closed form: for $t \geqslant t_0$,

$$x_t = \frac{x_{t_0} + z_{t_0}}{2} + \frac{x_{t_0} - z_{t_0}}{2}\exp\left(-2\sqrt{\frac{\mu}{L}}(t - t_0)\right)$$

$$= x_{t_0} + \frac{1}{2}\left(1 - \exp\left(-2\sqrt{\frac{\mu}{L}}(t - t_0)\right)\right)(z_{t_0} - x_{t_0})\,,$$

$$z_t = \frac{x_{t_0} + z_{t_0}}{2} + \frac{z_{t_0} - x_{t_0}}{2}\exp\left(-2\sqrt{\frac{\mu}{L}}(t - t_0)\right)$$

$$= z_{t_0} + \frac{1}{2}\left(1 - \exp\left(-2\sqrt{\frac{\mu}{L}}(t - t_0)\right)\right)(x_{t_0} - z_{t_0})\,.$$

In particular, taking $t_0 = T_k$, $t = T_{k+1}-$, we obtain $\tau_k = \tau_k'' = \frac{1}{2}\left(1 - \exp\left(-2\sqrt{\frac{\mu}{L}}(T_{k+1} - T_k)\right)\right)$ and thus $\tau_k' = \frac{\tau_k''}{1 - \tau_k} = \tanh\left(\sqrt{\frac{\mu}{L}}(T_{k+1} - T_k)\right)$. Finally, $\tilde{\gamma}_k = \gamma_{T_k} = \frac{1}{L}$ and $\tilde{\gamma}_k' = \gamma_{T_k}' = \frac{1}{\sqrt{\mu L}}$.

## F   Heuristic ODE scaling limit of the continuized acceleration

### F.1   Convex case

With the choices of parameters of Theorem [2].(1), the continuized acceleration is

$$\mathrm{d}x_t = \frac{2}{t}(z_t - x_t)\mathrm{d}t - \frac{1}{L}\nabla f(x_t)\mathrm{d}N(t)\,,$$

$$\mathrm{d}z_t = -\frac{t}{2L}\nabla f(x_t)\mathrm{d}N(t)\,.$$

The ODE scaling limit is obtained by taking the limit $L \to \infty$ (so that the stepsize $1/L$ vanishes) and rescaling the time $s = t/\sqrt{L}$. Some law of large number argument heuristically gives us that, as $L \to \infty$, $\mathrm{d}N(t) = \mathrm{d}N(\sqrt{L}s) \approx \sqrt{L}\mathrm{d}s$. Thus in the limit, we obtain

$$\mathrm{d}x_s = \frac{2}{\sqrt{L}s}(z_s - x_s)\sqrt{L}\mathrm{d}s - \frac{1}{L}\nabla f(x_s)\sqrt{L}\mathrm{d}s\,,$$

$$\mathrm{d}z_s = -\frac{\sqrt{L}s}{2L}\nabla f(x_s)\sqrt{L}\mathrm{d}s\,.$$

The second term of the first equation becomes negligible in the limit. Thus the equations simplify to

$$\frac{\mathrm{d}x_s}{\mathrm{d}s} = \frac{2}{s}(z_s - x_s)\,,$$

$$\frac{\mathrm{d}z_s}{\mathrm{d}s} = -\frac{s}{2}\nabla f(x_s)\,.$$

Thus

$$-\frac{s}{2}\nabla f(x_s) = \frac{\mathrm{d}z_s}{\mathrm{d}s} = \frac{\mathrm{d}}{\mathrm{d}s}\left(x_s + \frac{s}{2}\frac{\mathrm{d}x_s}{\mathrm{d}s}\right) = \frac{\mathrm{d}x_s}{\mathrm{d}s} + \frac{1}{2}\frac{\mathrm{d}x_s}{\mathrm{d}s} + \frac{s}{2}\frac{\mathrm{d}^2x_s}{\mathrm{d}s^2}\,,$$

and thus

$$\frac{\mathrm{d}^2x_s}{\mathrm{d}s^2} + \frac{3}{s}\frac{\mathrm{d}x_s}{\mathrm{d}s} + \nabla f(x_s) = 0\,.$$

This is the same limiting ODE as the one found by Su et al. [50] for Nesterov acceleration.

## F.2  Strongly-convex case

With the choices of parameters of Theorem 2.(2), the continuized acceleration is

$$\mathrm{d}x_t = \sqrt{\frac{\mu}{L}}(z_t - x_t)\mathrm{d}t - \frac{1}{L}\nabla f(x_t)\mathrm{d}N(t)\,,$$

$$\mathrm{d}z_t = \sqrt{\frac{\mu}{L}}(x_t - z_t)\mathrm{d}t - \frac{1}{\sqrt{\mu L}}\nabla f(x_t)\mathrm{d}N(t)\,.$$

Again, we take joint scaling $L \to \infty$, $s = t/\sqrt{L}$, with the approximation $\mathrm{d}N(t) \approx \sqrt{L}\mathrm{d}s$. We obtain

$$\mathrm{d}x_s = \sqrt{\frac{\mu}{L}}(z_s - x_s)\sqrt{L}\mathrm{d}s - \frac{1}{L}\nabla f(x_s)\sqrt{L}\mathrm{d}s\,,$$

$$\mathrm{d}z_s = \sqrt{\frac{\mu}{L}}(x_s - z_s)\sqrt{L}\mathrm{d}s - \frac{1}{\sqrt{\mu L}}\nabla f(x_s)\sqrt{L}\mathrm{d}s\,.$$

As before, the second term of the first equation becomes negligible in the limit. Thus the equations simplify to

$$\frac{\mathrm{d}x_s}{\mathrm{d}s} = \sqrt{\mu}(z_s - x_s)\,, \tag{44}$$

$$\frac{\mathrm{d}z_s}{\mathrm{d}s} = \sqrt{\mu}(x_s - z_s) - \frac{1}{\sqrt{\mu}}\nabla f(x_s)\,. \tag{45}$$

From (44), we have $z_s = x_s + \frac{1}{\sqrt{\mu}}\frac{\mathrm{d}x_s}{\mathrm{d}s}$, and by substitution in (45), we obtain

$$\frac{\mathrm{d}^2 x_s}{\mathrm{d}s^2} + 2\sqrt{\mu}\frac{\mathrm{d}x_s}{\mathrm{d}s} + \nabla f(x_s) = 0\,.$$

This is the so-called "low-resolution" ODE for Nesterov acceleration of Shi et al. [48].

## G   Continuized Accelerated Coordinate Descent with arbitrary sampling

In this section, we focus on the following problem:

$$\min_{x \in \mathbb{R}^d} f(x), \tag{46}$$

where $f$ is of the form $f : x \mapsto g(Rx)$ for some function $g$ and projector $R \in \mathbb{R}^{d \times d}$ (such that $R^2 = R$). We further assume that $f$ is smooth with respect to some matrix $M \in \mathbb{R}^{d \times d}$ and $\mu$-strongly convex with respect to $R$, *i.e.*:

$$\frac{\mu}{2}\|x - y\|_R^2 \leqslant f(x) - f(y) - \nabla f(x)^\top (x - y) \leqslant \frac{1}{2}\|x - y\|_M^2.$$

Note that $\mu$ can be equal to zero, but convergence will be slower in this case. We analyze the convergence of the following continuized coordinate descent iteration:

$$\mathrm{d}x_t = \eta_t(z_t - x_t)\mathrm{d}t - \gamma_t \int_\Xi \frac{R_{\xi\xi}}{\mathcal{P}_\xi}\nabla f(x_t, \xi)\mathrm{d}N(t, \xi)\,,$$
$$\mathrm{d}z_t = \eta_t'(x_t - z_t)\mathrm{d}t - \gamma_t' \int_\Xi \nabla f(x_t, \xi)\mathrm{d}N(t, \xi)\,, \tag{47}$$

where

$$\nabla f(x_t, \xi) = \frac{1}{\mathcal{P}_\xi}\nabla_\xi f(x_t), \tag{48}$$

with the coordinate gradient $\nabla_\xi f(x_t) = e_\xi e_\xi^\top \nabla f(x_t)$, with $e_\xi \in \mathbb{R}^d$ the unit vector associated with coordinate $\xi \in \{1, \ldots, d\}$ and $\mathcal{P}_\xi$ and $\mathrm{d}N$ are defined as in Section 6. Note that these iterations are slightly different from the previous stochastic gradient iteration since the stochastic gradient is not the same for $x_t$ and $z_t$ (same direction but different magnitudes). The following theorem is a continuized version of Hendrikx et al. [25], which is itself largely based on Nesterov and Stich [45].

**Theorem 9** (Continuized acceleration of coordinate descent). *Assume that the stochastic gradients are of the coordinate descent form* (48). *Besides, choose parameter L such that:*

$$L \geqslant \max_{\xi \in \Xi} \frac{M_{\xi\xi} R_{\xi\xi}}{\mathcal{P}_\xi^2} . \tag{49}$$

*Then the continuized acceleration* (60) *satisfies the following:*

*1. For $\eta_t = \frac{2}{t}, \eta_t' = 0, \gamma_t = \frac{1}{L}, \gamma_t' = \frac{t}{2L}$,*

$$\mathbb{E} f(x_t) - f(x_*) \leqslant \frac{2L \|z_0 - x_*\|_R^2}{t^2} .$$

*2. Assume further that $\mu > 0$ and choose the constant parameters $\eta_t = \eta_t' \equiv \sqrt{\frac{\mu}{L}}$, $\gamma_t \equiv \frac{1}{L}$, $\gamma_t' \equiv \frac{1}{\sqrt{\mu L}}$. Then,*

$$\mathbb{E} f(x_t) - f(x_*) \leqslant \left( f(x_0) - f(x_*) + \frac{\mu}{2} \|z_0 - x_*\|_R^2 \right) \exp \left( -\sqrt{\frac{\mu}{L}} t \right) .$$

*Proof.* Similarly to the proof in Appendix D.3, the proof of this theorem is along the same lines as the proof of Theorem 2, and we only highlight the major differences. The process $\bar{x}_t = (t, x_t, z_t)$ satisfies the equation

$$\mathrm{d}\bar{x}_t = b(\bar{x}_t)\mathrm{d}t + \int_\Xi G(\bar{x}_t, \xi)\mathrm{d}N(t, \xi), \quad b(\bar{x}_t) = \begin{pmatrix} 1 \\ \eta_t(z_t - x_t) \\ \eta_t'(x_t - z_t) \end{pmatrix}, \quad G(\bar{x}_t, \xi) = \begin{pmatrix} 0 \\ -\gamma_t \frac{R_{\xi\xi}}{\mathcal{P}_\xi} \nabla f(x_t, \xi) \\ -\gamma_t' \nabla f(x_t, \xi) \end{pmatrix} .$$

We also consider a slightly different Lyapunov function $\phi_t$ that takes into account the projector $R$:

$$\phi_t = A_t \left( f(x_t) - f_* \right) + \frac{B_t}{2} \|z_t - x_*\|_R^2$$

This change of norm is essential to take into account the fact that $f$ is not strongly convex with respect to the euclidean norm, but only with respect to $\|\cdot\|_R$. We apply Proposition 2 to $\phi_t = \varphi(\bar{x}_t) = \varphi(t, x_t, z_t)$ and obtain

$$\phi_t = \phi_0 + \int_0^t I_s \mathrm{d}s + M_t , \tag{50}$$

where $M_t$ is a martingale and

$$I_t = \langle \nabla \varphi(\bar{x}_t), b(\bar{x}_t) \rangle + \mathbb{E}_\xi \varphi(\bar{x}_t + G(\bar{x}_t, \xi)) - \varphi(\bar{x}_t) .$$

The computation of the first term remains the same: the inequality (29)-(30) holds. The computation of the second term becomes

$$\mathbb{E}_\xi \varphi(\bar{x}_t + G(\bar{x}_t, \xi)) - \varphi(\bar{x}_t) = A_t \left( \mathbb{E}_\xi f \left( x_t - \gamma_t \frac{R_{\xi\xi}}{\mathcal{P}_\xi} \nabla f(x_t, \xi) \right) - f(x_t) \right)$$

$$+ \frac{B_t}{2} \left( \mathbb{E}_\xi \|(z_t - x_*) - \gamma_t' \nabla f(x_t, \xi)\|_R^2 - \|z_t - x_*\|_R^2 \right) .$$

As $f$ is $M$-smooth,

$$f \left( x_t - \gamma_t \frac{R_{\xi\xi}}{\mathcal{P}_\xi} \nabla f(x_t, \xi) \right) - f(x_t) \leqslant \langle \nabla f(x_t), -\gamma_t \frac{R_{\xi\xi}}{\mathcal{P}_\xi} \nabla f(x_t, \xi) \rangle + \frac{1}{2} \|\gamma_t \frac{R_{\xi\xi}}{\mathcal{P}_\xi} \nabla f(x_t, \xi)\|_M^2 .$$

In the additive case, the variance is bounded by $\sigma^2$. In this case, we have that:

$$\|\frac{R_{\xi\xi}}{\mathcal{P}_\xi} \nabla f(x_t, \xi)\|_M^2 = \frac{M_{\xi\xi} R_{\xi\xi}}{\mathcal{P}_\xi^2} \|\nabla f(x_t, \xi)\|_R^2 \leqslant L \|\nabla f(x_t, \xi)\|_R^2, \tag{51}$$

and similarly:

$$\langle \nabla f(x_t), -\gamma_t \frac{R_{\xi\xi}}{\mathcal{P}_\xi} \nabla f(x_t, \xi) \rangle = -\gamma_t \frac{R_{\xi\xi}}{\mathcal{P}_\xi^2} \|\nabla_\xi f(x_t)\|^2 = \gamma_t \|\nabla f(x_t, \xi)\|_R^2. \tag{52}$$

Thus:

$$\mathbb{E}_\xi f\left(x_t - \gamma_t \frac{R_{\xi\xi}}{\mathcal{P}_\xi}\nabla f(x_t,\xi)\right) - f(x_t) \leqslant \gamma_t(1-\gamma_t L)\mathbb{E}_\xi\|\nabla f(x_t,\xi)\|_R^2.$$

Similarly, thanks to the unbiasedness of $\nabla f(x_t,\xi)$,

$$\mathbb{E}_\xi\|(z_t-x_*)-\gamma_t'\nabla f(x_t,\xi)\|_R^2 - \|z_t-x_*\|_R^2$$
$$= -2\gamma_t'\langle\mathbb{E}_\xi R\nabla f(x_t,\xi), z_t-x_*\rangle + \gamma_t'^2\mathbb{E}_\xi\|\nabla f(x_t,\xi)\|_R^2$$
$$\leqslant -2\gamma_t'\langle\nabla f(x_t), z_t-x_*\rangle + \gamma_t'^2\mathbb{E}_\xi\|\nabla f(x_t,\xi)\|_R^2.$$

This gives

$$\varphi(\bar{x}_t + G(\bar{x}_t)) - \varphi(\bar{x}_t) \leqslant -B_t\gamma_t'\langle\nabla f(x_t), z_t-x_*\rangle$$
$$+ \left(B_t\gamma_t'^2 - A_t\gamma_t(2-L\gamma_t)\right)\frac{1}{2}\mathbb{E}_\xi\|\nabla f(x_t,\xi)\|_R^2.$$

Combining the bounds, we obtain

$$I_t \leqslant \left(\frac{\mathrm{d}A_t}{\mathrm{d}t}-A_t\eta_t\right)\langle\nabla f(x_t), x_t-x_*\rangle + \left(\frac{\mathrm{d}B_t}{\mathrm{d}t}-B_t\eta_t'\right)\frac{1}{2}\|z_t-x_*\|_R^2$$
$$+ (A_t\eta_t - B_t\gamma_t')\langle\nabla f(x_t), z_t-x_*\rangle + \left(B_t\eta_t' - \frac{\mathrm{d}A_t}{\mathrm{d}t}\mu\right)\frac{1}{2}\|x_t-x_*\|_R^2$$
$$+ \left(B_t\gamma_t'^2 - A_t\gamma_t(2-L\gamma_t)\right)\frac{1}{2}\mathbb{E}_\xi\|\nabla f(x_t,\xi)\|_R^2.$$

We see that we obtain a result that is very similar to that of the deterministic case. The choices of parameters of Theorem 9 cancel all first five prefactors, and satisfy $\gamma_t = \frac{1}{L}$, $A_t L\gamma_t^2 = B_t\gamma_t'^2$. We thus obtain $I_t \leqslant 0$ and so $\phi_t$ is a supermartingale, and the rest follows as in Appendix D.2. $\square$

# H Accelerated Decentralized Optimization with Randomized Gossip Communications.

We now consider the setting of decentralized optimization considered in Section 7. More specifically, recall that we wish to solve:

$$\min_{x\in\mathbb{R}^d}\left\{f(x) = \frac{1}{|\mathcal{V}|}\sum_{v\in\mathcal{V}}f_v(x)\right\}, \tag{53}$$

where the function $f_v$ is privately held by node $v \in \mathcal{V}$. To solve this problem, a classical approach is to use a dual formulation [47, 25]. We first rewrite Problem (53) as:

$$\min_{X\in\mathbb{R}^{|\mathcal{V}|\times d},\ X_u=X_v\ \forall\{u,v\}\in\mathcal{E}}\left\{F(X) = \frac{1}{|\mathcal{V}|}\sum_{v\in\mathcal{V}}f_v(X_v)\right\}, \tag{54}$$

where $X_v \in \mathbb{R}^d$ corresponds to the local parameter of node $v$, and the equality constraints ensures equivalence between (53) and (54). The constraints are linear and can be expressed in matrix form as:

$$A^\top X = 0, \tag{55}$$

with $A \in \mathbb{R}^{\mathcal{E}\times\mathcal{V}}$ such that $\ker(A^\top) = \mathrm{Span}(1,...,1)$ the constant vector. The natural choice for matrix $A$ is to choose a square root of the Laplacian matrix of graph $G$. For $(e_v)_{v\in\mathcal{V}}$ and $(e_{\{v,w\}})_{\{v,w\}\in\mathbb{E}}$ the canonical bases of $\mathbb{R}^\mathcal{V}$ and $\mathbb{R}^\mathcal{E}$, $A$ is thus that for any $\{v,w\}\in\mathcal{E}$:

$$Ae_{\{v,w\}} = \sqrt{\mathcal{P}_{\{v,w\}}}(e_v - e_w).$$

Matrix $A$ then satisfies $AA^\top = \mathcal{L}$ the Laplacian matrix of graph $G$ with weights $\mathcal{P}_{\{v,w\}}$. Indeed, if $W_{\{v,w\}} = \mathcal{P}_{\{v,w\}}(e_v-e_w)(e_v-e_w)^\top$ corresponds to the gossip matrix for edge $\{v,w\}$, $A$ is such that:

$$AA^\top = \sum_{\{v,w\}\in\mathcal{E}} W_{\{v,w\}} = \mathcal{L}. \tag{56}$$

Then, introducing Lagrange multipliers $\lambda$, we obtain through Lagrangian duality that Problem (53) is equivalent to:

$$\max_{\lambda \in \mathbb{R}^{\mathcal{E} \times d}} -F^*(A\lambda), \tag{57}$$

with $F^*$ the convex conjugate of $F$. Following the approach of Hendrikx et al. [25], we then apply Accelerated Coordinate Descent to this dual problem. Yet, we use the *continuized* version of Theorem 9, which allows us to remove the global iterations counter on which previous approaches rely. We see that Problem (57) has exactly the right form to apply Theorem 9, leading to the following dual iterations:

$$d\lambda_t^{(y)} = \eta_t(\lambda_t^{(z)} - \lambda_t^{(y)})dt - \gamma_t \int_{\mathbb{R}_{\geqslant 0} \times \mathcal{E}} \frac{R_{\{v,w\}}}{\mathcal{P}_{\{v,w\}}^2} e_{\{v,w\}} e_{\{v,w\}}^\top A^\top \nabla F^*(A\lambda_t^{(y)}) dN(t, \{v,w\}),$$

$$d\lambda_t^{(z)} = \eta_t'(\lambda_t^{(y)} - \lambda_t^{(z)})dt - \gamma_t' \int_{\mathbb{R}_{\geqslant 0} \times \mathcal{E}} \frac{1}{\mathcal{P}_{\{v,w\}}} e_{\{v,w\}} e_{\{v,w\}}^\top A^\top \nabla F^*(A\lambda_t^{(y)}) dN(t, \{v,w\}),$$

$$\tag{58}$$

where $P = A^\dagger A$ with $A^\dagger$ is the pseudo-inverse of $A$, $R_{\{v,w\}} = e_{\{v,w\}}^\top A^\dagger A e_{\{v,w\}}$. Now, we multiply these iterations by $A$ on the left (which is standard), and we rewrite them with the following iterates:

$$y_t = A\lambda_t^{(y)}, \qquad z_t = A\lambda_t^{(z)}. \tag{59}$$

Note that $y_t, z_t \in \mathbb{R}^{|\mathcal{V}| \times d}$, and are thus variables associated with *nodes* of the graph.

$$dy_t = \eta_t(z_t - y_t)dt - \gamma_t \int_{\mathbb{R}_{\geqslant 0} \times \mathcal{E}} \frac{R_{\{v,w\}}}{\mathcal{P}_{\{v,w\}}^2} W_{\{v,w\}} \nabla F^*(y_t) dN(t, \{v,w\}),$$

$$dz_t = \eta_t'(y_t - z_t)dt - \gamma_t' \int_{\mathbb{R}_{\geqslant 0} \times \mathcal{E}} \frac{1}{\mathcal{P}_{\{v,w\}}} W_{\{v,w\}} \nabla F^*(y_t) dN(t, \{v,w\}),$$

$$\tag{60}$$

where we recall that $W_{\{v,w\}} = \mathcal{P}_{\{v,w\}}(e_v - e_w)(e_v - e_w)^\top$ corresponds to the gossip matrix for edge $\{v, w\}$. Besides, the dual gradients $\nabla F^*(y_t)$ are such that $e_v^\top \nabla F^*(y_t) = \nabla f_v^*(e_v^\top y_t)$, and so each component can be computed locally at node $v$.

In summary, the distributed decentralized algorithm writes as follows. Upon activation of edge $\{v_k, w_k\}$ at time $T_k$,

$$G_{\{v_k,w_k\}}(T_k) = \omega_{\{v_k,w_k\}} \left[ \nabla f^*((y_{T_k^-})_{v_k}) - \nabla f^*((y_{T_k^-})_{w_k}) \right]$$

$$y_{T_k}(v_k) = y_{T_k^-}(v_k) - \gamma_t \frac{R_{\{v_k,w_k\}}}{\mathcal{P}_{\{v_k,w_k\}}^2} G_{\{v_k,w_k\}}(T_k),$$

$$y_{T_k}(w_k) = y_{T_k^-}(w_k) + \gamma_t \frac{R_{\{v_k,w_k\}}}{\mathcal{P}_{\{v_k,w_k\}}^2} G_{\{v_k,w_k\}}(T_k), \tag{61}$$

$$z_{T_k}(v_k) = z_{T_k^-}(v_k) - \gamma_t' G_{\{v_k,w_k\}}(T_k),$$

$$z_{T_k}(w_k) = z_{T_k^-}(w_k) + \gamma_t' G_{\{v_k,w_k\}}(T_k).$$

Between these updates, $y_t(v)$ and $z_t(v)$ locally mix at all nodes $v \in \mathcal{V}$, according to the coupled ODE:

$$dy_t(v) = \eta_t(z_t(v) - y_t(v))dt,$$

$$dz_t(v) = \eta_t'(y_t(v) - z_t(v))dt.$$

This algorithm can be implemented with local computations and pairwise communications only, since an update along edge $\{v, w\}$ only involves the parameters and functions of nodes $v$ and $w$. In order to fully describe this algorithm, we need to specify the various parameters. We do so, with the corresponding rate of convergence, in the following theorem.

**Theorem 10** (Asynchronous Accelerated Decentralized Optimization). *Assume that each $f_v$ is $\mu$-strongly-convex with $\mu > 0$ and $L$-smooth. Let $L_{\mathrm{dual}} = \frac{1}{\mu} \max_{\{v,w\}} \frac{R_{\{v,w\}}}{\mathcal{P}_{\{v,w\}}}$, where we recall*

*that $R_{\{v,w\}} = (A^\dagger A)_{\{v,w\},\{v,w\}}$. Then, let $\theta'_{\mathrm{ARG}} = \sqrt{\mu_{\mathrm{gossip}} / \max_{\{v,w\}} \frac{R_{\{v,w\}}}{\mathcal{P}_{\{v,w\}}}}$ where $\mu_{\mathrm{gossip}}$ is the smallest non-zero eigenvalue of the Laplacian of the graph $\mathcal{G}$, and $\kappa = L/\mu$ is a bound on the condition number of $f$. We choose the constant parameters $\eta_t = \eta'_t \equiv \frac{\theta'_{\mathrm{ARG}}}{\sqrt{\kappa}}$, $\gamma_t \equiv \frac{1}{L_{\mathrm{dual}}}$, $\gamma'_t \equiv \sqrt{\frac{L}{\mu_{\mathrm{gossip}} L_{\mathrm{dual}}}}$. The iterates produced by the algorithm described in* (61) *verify:*

$$\mathbb{E} \sum_{v \in V} \frac{1}{2} \|\nabla f_v^*(z_t(v)) - x_\star\|^2 \leqslant C_0^{\mathrm{dual}} \exp\left(-\frac{\theta'_{\mathrm{ARG}}}{\sqrt{\kappa}} t\right),$$

*with $C_0^{\mathrm{dual}} = \frac{\lambda_{\max}(AA^\top)}{\mu} \left( F^*(A\lambda_0^{(y)}) - F^*(A\lambda_\star) + \frac{\mu_{\mathrm{gossip}}}{2L} \|\lambda_0^{(z)} - \lambda_\star\|_{A^\dagger A}^2 \right)$, with $\lambda_\star$ a solution to the dual problem.*

Note that $\theta'_{\mathrm{ARG}}$ is slightly different from $\theta_{\mathrm{ARG}}$. Yet, following Hendrikx et al. [25], an equivalent of Corollary 1 can be obtained for $\theta'_{\mathrm{ARG}}$. To obtain Theorem 6, we simply choose $\lambda_0^{(y)} = \lambda_0^{(z)}$ and bound the dual function suboptimality by the distance to optim using the smoothness and strong convexity of $F^*$.

We stress the fact that the accelerated algorithm described in this section, as well as accelerated randomized gossip in Section 6, are decentralized and asynchronous: operations are local and do not require any global synchronization, provided that a continuous time clock can be shared. This is possible only thanks to the continuized framework. However, there are some limitations: even if these algorithms are the first to achieve these rates without any global synchronization, computations and communications are here assumed to happen instantly, or to take a negligible time. Handling communication and computation physical capacity constraints such as delays or node/edge overloads in our algorithms as in [23] combined with accelerated schemes is left for future works.

*Proof.* First note that the Hessian of the dual objective writes for some $\lambda \in \mathbb{R}^{|\mathcal{E}| \times d}$:

$$A^\top \nabla^2 F^*(A\lambda)A \succcurlyeq \frac{1}{L} A^\top A, \tag{62}$$

since $F^*$ is $L^{-1}$ strongly-convex when $F$ is $L$-smooth [31]. Thus, the dual objective is $\mu_{\mathrm{gossip}}/L$ strongly convex on the orthogonal of the kernel of $A$. Similarly, the smoothness of the dual objective in direction $\{v, w\}$ is equal to:

$$M_{\{v,w\}\{v,w\}} = e_{\{v,w\}}^\top A^\top \nabla^2 F^*(A\lambda) A e_{\{v,w\}} \preccurlyeq \frac{1}{\mu} e_{\{v,w\}}^\top A^\top A e_{\{v,w\}} = \frac{\mathcal{P}_{\{v,w\}}}{2\mu}. \tag{63}$$

Thus, we have that:

$$L_{\mathrm{dual}} = \max_{\{v,w\}} \frac{M_{\{v,w\}\{v,w\}} R_{\{v,w\}}}{\mathcal{P}_{\{v,w\}}^2} = \frac{1}{\mu} \max_{\{v,w\}} \frac{R_{\{v,w\}}}{\mathcal{P}_{\{v,w\}}}. \tag{64}$$

Then, the result follows directly from applying Theorem (9), together with the smoothness of the dual gradients, since:

$$\mathbb{E} \sum_{v \in V} \frac{1}{2} \|\nabla f_v^*(z_t(v)) - x_\star\|^2 \leqslant \mathbb{E} \frac{1}{2\mu} \|A\lambda_t^{(z)} - A\lambda_\star\|^2 \leqslant \frac{\lambda_{\max}(AA^\top)}{2\mu} \mathbb{E} \|\lambda_t^{(z)} - \lambda_\star\|_R^2. \tag{65}$$

$\qquad\square$

Note that the primal parameter that we are interested in is $x_t = \nabla f^*(z_t)$, and not $y_t$ or $z_t$ which are dual parameters.