# OpenReview forum: "Continuized Accelerations of Deterministic and Stochastic Gradient Descents, and of Gossip Algorithms"
_NeurIPS.cc/2021/Conference — NeurIPS 2021 Oral_

### Official Review · Reviewer_ZD1t · 2021-06-28

**Rating:** 7
**Confidence:** 3

**Summary:**

This paper introduces a new continuous-time framework for understanding Nesterov’s accelerated gradient descent. The authors consider the following dynamics for two variables (x, z): at the jump times of a Poisson process, each of the two variables x and z takes a step in the negative gradient of the objective function. In between these jumps, the variables x and z mix continuously according to an ODE. The authors show that with appropriate choices for the parameters, this process exhibits similar guarantees as Nesterov’s algorithm.

Continuous-time versions of Nesterov’s algorithm have been considered before, but usually from the perspective of letting the step size of the algorithm tend to zero and thus recovering an ODE. Although it is usually much simpler to study the resulting ODE, it is quite non-trivial to start with an ODE and find a suitable discretization that preserves its convergence guarantees; thus, many of these prior works simply reverse-engineer Nesterov’s algorithm without providing an understanding of how to produce discretized accelerated algorithms. The main novelty in the present approach is that there is no need to discretize the continuous-time dynamics; the gradient steps already happen at discrete times (the jumps of a Poisson process) which can be simulated (with the caveat that the algorithm is now randomized). In between the jumps, the pair (x, z) evolves according to an ODE which does not depend on the objective function (besides the smoothness/convexity parameters). For the parameter choices leading to accelerated rates for convex-smooth and strongly convex-smooth classes, the authors show that in fact this ODE can be solved in closed form. Even if this were not the case, it is noteworthy that this ODE can be implemented in the usual oracle model of optimization.

Besides introducing this framework, the authors provide extensions to stochastic gradients with “additive” or “multiplicative” noise, as well as applications to accelerated gossip.

**Limitations And Societal Impact:**

Yes.

**Main Review:**

Despite the immense amount of prior work attempting to decipher Nesterov’s algorithm, I believe this current work is novel and interesting. Indeed, passing from the continuous-time framework to an implementable algorithm is much more straightforward, making this a promising method for bridging continuous-time analysis with optimization algorithms.

Specific Comments:
- In Thm. 6, one of the two expectation symbols seems redundant.
- There is an extraneous factor of 2 in eq. (36).

**Time Spent Reviewing:**

1

---

> ### Author Response · Authors · 2021-08-09
> **author response to reviewer ZD1t**
>
> We thank the reviewer for his work and his encouraging and positive comments. The summary is very accurate and we are happy to share the same opinion about the relation with previous ODE-based interpretations of Nesterov's method.
>
> We corrected the typos and included a series of smaller improvements to style and clarity.

---

> > ### Comment · Reviewer_ZD1t · 2021-08-12
> > **Acknowledged**
> >
> > Thank you for the response. I have read the other reviews and responses and my initial opinion still stands.

---

### Official Review · Reviewer_ges3 · 2021-07-07

**Rating:** 9
**Confidence:** 4

**Summary:**

This paper studies accelerated gradient descent algorithm for convex optimization from a continuous-time perspective. The authors introduce the continuized version of Nesterov's accelerated gradient descent (AGD) algorithm, which takes gradient steps at random times with exponentially distributed intervals. The authors show the continuized AGD has accelerated convergence rates in expected function values similar to the standard Nesterov's AGD. The authors derive an exact discretization of the continuized AGD which resembles Nesterov's AGD with random parameters, and show the algorithm has accelerated convergence rates similar to the rates in continuous time. The authors extend their algorithms and analysis to stochastic gradient descent with additive and multiplicative noise. The authors apply their framework to derive accelerated randomized gossip algorithm which can be implemented in a decentralized way.

**Limitations And Societal Impact:**

The authors have addressed the limitations of their work.

**Main Review:**

This paper introduces the continuized version of Nesterov's accelerated gradient descent (AGD) which takes gradient steps at random times with exponentially distributed intervals. Equivalently, the continuized AGD is modeled as a system of stochastic differential equations driven by Poisson noise. This is a novel perspective that is different from previous works on continuous-time acceleration, which take the limit of step size going to 0 to recover second-order differential equations (or a pair of first-order ODEs). The continuized version of AGD introduced in this work has many advantages, as described in this paper. The continuized AGD in continuous time has accelerated convergence rates similar to the rates for AGD. Furthermore, the continuized AGD can also be discretized exactly (whereas typical discretization of ODEs have errors that need to be controlled carefully), and the resulting algorithm looks like AGD with random parameters with similar convergence rates. The authors extend their algorithms and analysis to stochastic gradient descent, and they also apply their framework to accelerate decentralized randomized gossip algorithm.

This is an excellent paper that makes a novel conceptual contribution to the study of accelerated gradient descent algorithm. The authors propose a new continuous-time perspective of AGD which will likely influence future work in this area. This paper is well-written, with a good review of previous works and background materials, and a clean presentation of results. The authors prove strong theoretical results in a variety of settings, and the proofs seem rigorous.

**Time Spent Reviewing:**

5

---

> ### Author Response · Authors · 2021-08-09
> **author response to reviewer ges3**
>
> The summary of the paper is very accurate and we were very happy to read the reviewer's enthusiastic opinion. We thank the reviewer for his positive and very encouraging review, as well as for his time and careful reading.

---

### Official Review · Reviewer_gPzr · 2021-07-09

**Rating:** 7
**Confidence:** 5

**Summary:**

The authors introduce a novel idea to analyze accelerated methods from a continuized point of view, which allows to have a type of "random learning rates". In particular, this allows to be oblivious of the iteration in an asynchronous network, as long as you have a common synchronized clock for the nodes. The analysis shows accelerated rates for their deterministic and stochastic methods, one of which can be applied to obtain acceleration in an asynchronous gossip algorithm.

**Limitations And Societal Impact:**

--

**Main Review:**

This is a good paper. The idea of using a continuized analysis for accelerated methods is quite original and it is shown to have some applications that improve, in some domains, the previous state of the art. In my opinion, part of the magic is on Proposition 2, that allows to ignore a term when checking that the Lyapunov functions are supermartingales (otherwise one could just do the regular analysis of accelerated methods, that usually only sets the parameters' values at the end of it, and one could consider which random variables they need to be in order for a final inequality to be satisfied. Because authors use proposition 2, the analysis becomes quite clean and simple).

The experiments are good showcases of the theory and the examples given as applications are a good addition to check that this analysis gets some gains with respect to non-continuized approaches.

I would have liked to see a comment on the synchronized clocks and the implications of working with them (not how to implement the clocks). In accelerated randomized gossip it is perfectly clear that each step happens when a communication step happens and that in that case the clock is used to set the parameters of the mixing of the two local variables.

However in the other cases, like that of acceleration of asynchronous decentralized optimization, a gradient step only happens "when the clock ticks", which is something that depends on the communication. This is as opposed to locally computing gradients as fast as your local processor can. So it seems that in these other cases that the paper describes the local nodes would be idle (without computing any gradients) until a communication step happens, which is not realistic. Please clarify to me if I am missing something and if this is not the case.

Still, I think the theoretical contributions of this paper and the other applications are enough for this paper to be published at Neurips.


Minor comments:

l213 rephrase

Figure 2: runes -> runs

Post-rebuttal: I thank the authors for their detailed response and willingness to add a comment on the idleness of the nodes due to the setting and the clock


**Time Spent Reviewing:**

13

---

> ### Author Response · Authors · 2021-08-09
> **author response to reviewer gPzr**
>
> We thank the reviewer for the time he spent on the review, his meticulous reading, and his positive and very encouraging comments.
>
> As correctly emphasized by the reviewer, it seems a bit strange to keep all nodes idle between communication steps. Perhaps surprisingly, the method already achieves the lower complexity bounds for decentralized optimization ($O(\sqrt{\kappa}\log(1/\varepsilon))$ gradient computations and $O(\sqrt{\kappa / \gamma}\log(1/ \varepsilon))$ communication steps, where $\kappa$ is the condition number and $\gamma$ the eigengap of the graph; see http://proceedings.mlr.press/v70/scaman17a/scaman17a.pdf), so there would therefore be no apparent theoretical incentive for considering such additional computations.
>
> That being said, such additional gradient computations would certainly improve the practical effectiveness of the method. However, more practical analyses would also require incorporating computation delays into the analysis, accounting for the fact that gradient computations might not be accurate (i.e., computed at old iterates). From what we can tell, incorporating such delays would quickly complicate the analyses which we wanted to keep as modular as possible.
>
> As suggested by the reviewer, we will add a short discussion on this topic in the paper. The current modifications also contain a series of smaller improvements to style and clarity.

---

### Official Review · Reviewer_v413 · 2021-07-20

**Rating:** 7
**Confidence:** 4

**Summary:**

This paper introduces a "continuized" randomized variant of the Nesterov acceleration scheme for convex optimization. In particular, "updates" are triggered at random i.i.d. exponential intervals of rate 1, leading to Markovian dynamics. The implications of this scheme are investigated in terms of streamlining, simplifying and improving the Nesterov acceleration.

**Main Review:**

I found the paper interesting. It may be a variant of the Nesterov scheme, but it introduces new useful insights. In looking though the proofs, it appears that one key point is the randomized triggering of "updates" at random i.i.d. exponential intervals of rate 1, using Poisson measures that lead to Markovian dynamics. I am wondering whether the same effect could be obtained in a discretized fashion, that is, flipping i.i.d. coins to trigger the updates. A similar Markovian structure would emerge, but in discrete time now. I think it would be useful for the authors to address this point, because it arises naturally in the reader's mind.

**Time Spent Reviewing:**

4

---

> ### Author Response · Authors · 2021-08-09
> **Author response to reviewer v413**
>
> We thank the reviewer for the time spent on the review, his careful reading, and his very inspired comments.
>
> We believe that the remark on the discrete version of the randomized process is a very good question and that it actually works. It actually corresponds to our starting point for creating the method. In fact, the setting proposed by the reviewer can be seen as an interpolation between the classical and the continuized setting. Indeed, flipping coins (with time being discrete) would mean having geometric random variables between two gradient steps instead of exponential ones. That is, denoting by $p$ the probability to trigger an update at a discrete time step: (i) if $p=1$, this is the classical setting, while (ii) if $p \to 0$ while renormalizing time, we obtain the continuized framework.
>
> The exposition is simpler in the $p \to 0$ limit, but we thank the reviewer for pointing out this interpolation, which we will mention as a remark in the final version.

---

### Decision · Program_Chairs · 2021-09-27

**Decision:**

Accept (Oral)

**Comment:**


A fine paper that provides an original and refreshing perspective on one of the most enigmatic algorithmic cornerstones in convex optimization, namely, accelerated gradient descent.